# EMBEDDING-BASED CONTEXT-AWARE RERANKER

**Ye Yuan[1,2]\*, Mohammad Amin Shabani[3], Siqi Liu[3]**
[1]McGill University, [2]Mila - Quebec AI Institute, [3]RBC Borealis

## ABSTRACT

Retrieval-Augmented Generation (RAG) systems rely on retrieving relevant evidence from a corpus to support downstream generation. The common practice of splitting a long document into multiple shorter passages enables finer-grained and targeted information retrieval. However, it also introduces challenges when a correct retrieval would require inference across passages, such as resolving coreference, disambiguating entities, and aggregating evidence scattered across multiple sources. Many state-of-the-art (SOTA) reranking methods, despite utilizing powerful large pretrained language models with potentially high inference costs, still neglect the aforementioned challenges. Therefore, we propose *Embedding-Based Context-Aware Reranker* (**EBCAR**), a lightweight reranking framework operating directly on embeddings of retrieved passages with enhanced cross-passage understandings through the structural information of the passages and a hybrid attention mechanism, which captures both high-level interactions across documents and low-level relationships within each document. We evaluate EBCAR against SOTA rerankers on the ConTEB benchmark, demonstrating its effectiveness for information retrieval requiring cross-passage inference and its advantages in both accuracy and efficiency. Our source code is available at https://github.com/BorealisAI/EBCAR.

## 1 INTRODUCTION

Retrieval-Augmented Generation (RAG) systems (Lewis et al., 2020; Wu et al., 2024) have become a cornerstone for enabling language models to incorporate external knowledge in complex reasoning tasks such as question answering (Mao et al., 2021; Xu et al., 2024b), fact verification (Adjali, 2024; Yue et al., 2024), and dialogue generation (Huang et al., 2023; Wang et al., 2024a). A typical RAG pipeline consists of a retriever, identifying relevant passages from a large corpus, and a reranker, reordering these candidates to surface the most useful evidence for downstream generation. The reranker plays a critical role in filtering out noisy retrieval results and promoting passages that are more faithful, complete, and relevant to the input query (Glass et al., 2022; de Souza P. Moreira et al., 2024). To support fine-grained retrieval, modern pipelines often rely on passage-level indexing, where a long document is split into shorter fixed-size passages for retrieval. This chunking process refines the granularity of retrieval and improves the readability of retrieved content for downstream models (Xu et al., 2024a; Jiang et al., 2024). While long-context encoders and embedding models have recently emerged (Zhang et al., 2024; Warner et al., 2025; Boizard et al., 2025; Zhu et al., 2024), passage-level retrieval remains the dominant design choice in many deployed systems due to its effectiveness and efficiency (Wu et al., 2024; Conti et al., 2025).

Existing reranking methods typically fall into three paradigms (Sharifymoghaddam et al., 2025): **(i)** Pointwise scoring models evaluate each query-passage pair independently. For example, Nogueira et al. (2019) and Nogueira et al. (2020) score individual query-passage pairs using pretrained language models (PLMs), by formulating the task as a binary classification problem or as a generative task requiring a "True" or "False" token as the output. **(ii)** Pairwise approaches compare pairs of passages to infer relative relevance; for instance, Pradeep et al. (2021) estimates the probability that one passage is more relevant than another given the query. **(iii)** Listwise methods, in contrast, operate over the entire set of retrieved candidates at once. One common practice is to prompt large language models (LLMs) to directly output a reranked list (Sun et al., 2023). Another is to distill supervision signals from LLMs into smaller generative models (Pradeep et al., 2023a;b; Gangi Reddy

---

*Work done while doing an internship at RBC Borealis. Corresponding to ye.yuan3@mail.mcgill.ca.

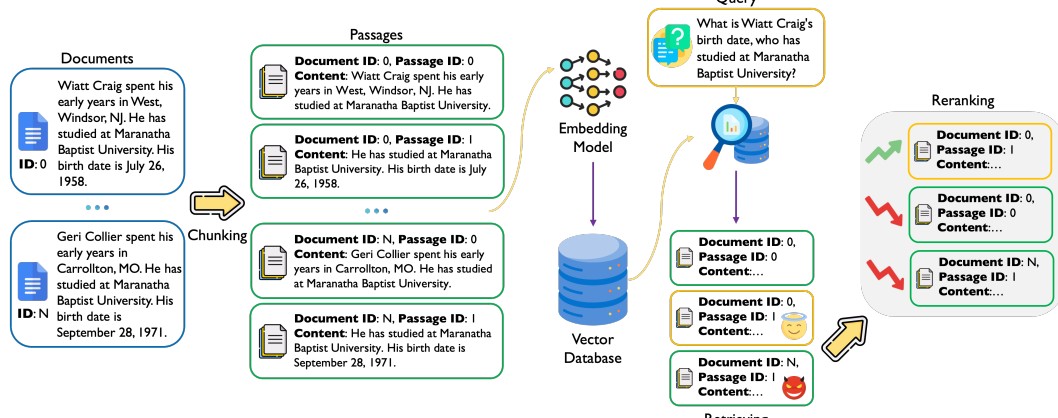

Figure 1: A Retrieval-Augmented Generation (RAG) pipeline with passage-level retrieval. Long documents are chunked into passages before being embedded into a dense vector store. At query time, top-$k$ passages are retrieved and reranked to provide evidence for downstream generation.

et al., 2024). Recently, inference-time methods have been proposed that prompt an LLM to compute calibrated attention-based scores across the retrieved passages (Chen et al., 2025).

Despite recent progress, two important challenges in reranking remain underexplored. First, across pointwise, pairwise, and listwise paradigms, most rerankers rely on feeding the raw text of retrieved passages and the query into large pretrained language models (PLMs) for scoring. This strategy incurs substantial computational cost and latency during inference, making it less practical for real-world deployments. Second, many existing methods are evaluated on idealized benchmarks that assume the necessary information for answering a query is fully contained within a single passage, leaving the capabilities of current reranking strategies underexplored in scenarios that demand cross-passage inference (Conti et al., 2025; Thakur et al., 2025; Zhou et al., 2025). As illustrated in Figure 1, consider the query "What is Wiatt Craig's birth date, who has studied at Maranatha Baptist University?" Passages related to "Wiatt Craig", "birth date", and "Maranatha Baptist University" may all be retrieved individually. However, multiple retrieved passages mention the birth date of a person who studied at Maranatha Baptist University, but without resolving the pronoun "he", the reranker cannot determine which passage truly pertains to "Wiatt Craig", leading to misranking.

To address the dual challenges of inference inefficiency and insufficient modeling of cross-passage context, we propose *Embedding-Based Context-Aware Reranker* (**EBCAR**). In contrast to prior rerankers that rely on scoring raw text using large PLMs, EBCAR operates directly on the dense embeddings of queries and retrieved passages. These passage embeddings are readily available, as they are stored in a vector database during indexing. The query embedding is obtained at inference time using the same encoder employed during passage indexing, ensuring compatibility in representation space. This design decouples reranking from costly text-to-text inference, substantially improving computational efficiency and enabling low-latency deployment. To facilitate cross-passage inference, EBCAR incorporates structural signals, such as document IDs and the positions of passages within their original documents, via positional encodings. These encodings are processed by a transformer encoder equipped with a *hybrid attention mechanism*, comprising two complementary multi-head attention modules: a *shared full attention* module and a *dedicated masked attention* module. Both modules operate on the same input representations but differ in their attention scopes. The shared full attention allows the query and all passage embeddings to attend to one another, capturing global interactions across retrieved passages. In contrast, the dedicated masked attention restricts attention scopes to the passages originating from the same documents, as determined by their document IDs. This hybrid attention design enables the model to infer over both inter-document context and fine-grained intra-document dependencies, promoting more accurate evidence aggregation and entity disambiguation. We evaluate EBCAR on ConTEB (Conti et al., 2025), a challenging benchmark originally designed to test retrieval models' use of document-wide context but used in our work to assess rerankers. Our experiments include both in-distribution evaluations and out-of-distribution zero-shot tests on entirely unseen domains, where EBCAR achieves promising performance.

Our main contributions are summarized as follows:

- We highlight the challenges of both efficiency and cross-passage inference in reranking, and introduce a lightweight embedding-based framework to addresses them.

- We propose a hybrid attention architecture that incorporates structural signals, such as document identity and passage position, through positional encodings, and combines shared full and dedicated masked attention modules to enhance context-aware reranking.

- We conduct extensive evaluations on ConTEB, demonstrating that EBCAR achieves promising performance in both in-distribution and out-of-distribution zero-shot settings, with significantly lower inference latency.

## 2 RELATED WORK

In this section, we review prior work related to reranking from three perspectives: existing context-aware methods to enhance retrieval systems, the design of reranking methods, and the evaluation protocols used to assess their capabilities.

**Contextual Retrieval.** Prior work has proposed several methods to enhance retrieval systems with broader contextual understanding beyond isolated passages. For instance, late chunking (Günther et al., 2025), semantic chunking (Qu et al., 2025), and hierarchical retrieval (Arivazhagan et al., 2023; Choe et al., 2025) aim to improve retrieval granularity by better aligning the unit of retrieval with semantic boundaries or document structure. These methods operate at the retrieval or generation stage and have proven effective in enhancing RAG pipelines. In contrast, our work focuses on the reranking stage and introduces architectural mechanisms to explicitly model intra and inter-passage dependencies within the reranker. We position EBCAR as complementary to the above techniques: it can be integrated seamlessly into RAG pipelines that already incorporate context-aware retrieval or generation strategies, providing an additional layer of cross-passage inference during candidate reranking.

**Reranking Methods.** Existing reranking approaches can be broadly categorized into three paradigms: pointwise, pairwise, and listwise. **(i)** Pointwise models treat reranking as a binary classification or regression task and score each query–passage pair independently. MonoBERT (Nogueira et al., 2019) formulates reranking as a binary classification problem using BERT to predict whether a passage is relevant to a given query. MonoT5 (Nogueira et al., 2020) adopts a generative approach, fine-tuning a T5 model to generate the tokens "True" or "False" and computing the predicted relevance score via softmax over the logits of these two tokens. However, pointwise rerankers incur high inference costs, as they require a separate forward pass for each candidate passage.

**(ii)** Pairwise methods compare two candidate passages at a time to infer their relative relevance to the query. DuoT5 (Pradeep et al., 2021), for instance, estimates the probability that one passage is more relevant than another. The resulting pairwise orders are typically aggregated, using sorting heuristics or tournament-based voting, to yield a final ranking. While pairwise approaches can better capture comparative relevance than pointwise methods, their computational cost grows quadratically with the number of candidates.

**(iii)** Listwise methods score or reorder the entire set of retrieved passages jointly. While earlier methods treated this holistically using traditional architectures, most recent listwise approaches leverage large language models (LLMs) in different ways, leading to three main subcategories in practice. *Prompt-only reranking* directly queries a powerful LLM, e.g., GPT-3.5 or GPT-4, to return a ranked list of passages in a single forward pass. For example, RankGPT (Sun et al., 2023) prompts the model to reorder passages based on their relevance to the query. While effective in zero-shot settings, this approach depends on proprietary APIs and incurs high inference latency and computational cost. *LLM-supervised distillation* extracts listwise supervision signals from prompt-only LLMs and uses them to fine-tune smaller, open-source models. RankVicuna (Pradeep et al., 2023a), RankZephyr (Pradeep et al., 2023b), and Lit5Distill (Tamber et al., 2023) fine-tune 7B-scale or smaller models using the ranked outputs from GPT-style models. Similarly, FIRST (Gangi Reddy et al., 2024) and FIRST-Mistral (Chen et al., 2024) use the logits of the first generated token as a ranking signal, while PE-rank (Liu et al., 2025) compresses passage context into a single embedding for efficient listwise reranking. These distilled models preserve much of the performance of larger LLMs while offering improved inference efficiency. *Inference-time relevance extraction* bypasses

generation entirely by computing relevance scores from internal attention or representation patterns of LLMs. ICR (Chen et al., 2025), for instance, derives passage-level relevance using attention-based signals from a calibration prompt and two forward passes through the LLM. While more efficient, these methods still rely on large models and full-text processing during inference.

**Evaluation Settings.** Most existing rerankers are trained on the MS-MARCO passage ranking dataset (Bajaj et al., 2016) and evaluated in a zero-shot manner on benchmarks such as the TREC Deep Learning (DL) tracks (Craswell et al., 2020; 2021) and the BEIR benchmark suite (Thakur et al., 2021). While these datasets have driven substantial progress, they predominantly feature self-contained passages that independently provide sufficient information to answer a query. This design simplifies the reranking task and limits the need for inference across multiple passages or leveraging broader document-level context (Conti et al., 2025; Thakur et al., 2025). In addition, structural signals such as document identifiers or passage positions are often absent or unused in these benchmarks. To address these limitations, we evaluate our proposed EBCAR on ConTEB (Conti et al., 2025), a benchmark originally developed to assess retrievers' use of document-wide context, which we use as a challenging testbed to evaluate rerankers' ability to infer across multiple retrieved passages in both in-distribution and out-of-distribution scenarios. In this work, we use the term *cross-passage inference* to broadly refer to settings where relevant information is distributed across multiple passages, requiring models to aggregate and reconcile evidence across them.

## 3 METHODOLOGY

Rerankers aim to refine a list of candidate passages retrieved for a query, producing a final ranking that better reflects passage relevance. In this work, we propose an embedding-based reranking framework that jointly scores candidate passages while leveraging cross-passage context and document-level structure. Our approach operates entirely in the embedding space, enabling fast and scalable inference. Below, we formally define the embedding-based reranking problem and present our proposed method in detail.

### 3.1 EMBEDDING-BASED FRAMEWORK

Given a user-issued query $x_q$ and a set of candidate passages $\{x_1, x_2, \ldots, x_k\}$ retrieved from a vector store, the goal is to reorder them based on their contextual relevance to the query. Each passage $x_i$ is associated with a pre-computed embedding $p_i \in \mathbb{R}^d$, obtained during indexing via a shared embedding function $\mathcal{E}(\cdot)$, i.e., $p_i = \mathcal{E}(x_i)$. These passage embeddings remain fixed during inference, and only the query needs to be encoded as $q = \mathcal{E}(x_q)$ using the same embedding function. The reranker aims to produce a permutation $\pi(\cdot)$ over the indices $\{1, 2, \ldots, k\}$ such that the reordered list $\{x_{\pi(1)}, \ldots, x_{\pi(k)}\}$ places the most relevant passages to $q$ at the top. This formulation enables efficient and parallelizable inference over dense embeddings.

### 3.2 EMBEDDING-BASED CONTEXT-AWARE RERANKER

**Architecture.** To capture both inter-document and intra-document dependencies among candidate passages, we propose a hybrid-attention-based reranking architecture, EBCAR, that operates on pre-computed dense embeddings. As shown in Figure 2, EBCAR integrates document identifiers and passage position information into the encoding process, and leverages a hybrid attention mechanism to model both global and document-local interactions.

We begin by augmenting the passage embeddings with structural signals. Given $k$ candidate passages $\{x_1, \ldots, x_k\}$ with corresponding dense embeddings $\{p_1, \ldots, p_k\}$, each passage embedding $p_i \in \mathbb{R}^d$ is enriched with two additional embeddings: a *document ID embedding* $\text{doc}(i) \in \mathbb{R}^d$, indicating the document to which $x_i$ belongs, and a *passage position encoding* $\text{pos}(i) \in \mathbb{R}^d$, reflecting the position of $x_i$ within its original document. These components are combined as $\tilde{p}_i = p_i + \text{doc}(i) + \text{pos}(i)$. The document ID embeddings are only used to distinguish the unique documents that the current retrieved passages belong to, instead of all the documents in the corpus. The $i$-th document ID embedding corresponds to the $i$-th unique document from the current $k$ retrieved passages. As a result, the document ID embedding table has a maximum size of $k \times d$ and remains fixed across training and inference. Importantly, the document ID embeddings are *relative*

*and local* to each retrieved candidate set, rather than global corpus-level identifiers. At inference time, we re-index document IDs on-the-fly according to the unique source documents within the $k$ retrieved passages. The document ID embeddings table is reused across all queries and kept *frozen*. For example, suppose the retriever returns $k = 6$ passages for query $q$, where $x_1, x_2, x_4$ come from document A, $x_3, x_5$ from document B, and $x_6$ from document C. EBCAR dynamically assigns three document IDs for this candidate set, e.g., $\text{doc}(1) = \text{doc}(2) = \text{doc}(4) = \mathbf{e}_1$, $\text{doc}(3) = \text{doc}(5) = \mathbf{e}_2$, $\text{doc}(6) = \mathbf{e}_3$, where $\mathbf{e}_1$, $\mathbf{e}_2$, and $\mathbf{e}_3$ are selected from the same fixed-size embedding table. When processing another query $q'$, the retriever may return a different set of six passages $x'_1, \ldots, x'_6$ from entirely different documents, e.g., A', B', C'; these will again be assigned document embeddings $\mathbf{e}_1, \mathbf{e}_2, \mathbf{e}_3$ from the same table. The primary purpose of this design is to enable the model to learn to recognize which passages originate from the same document through these relative document ID embeddings, thereby facilitating within-document reasoning. As a secondary benefit, this dynamic formulation also allows new documents to be incorporated without retraining: each new document is simply assigned a temporary local document ID during reranking. The passage position encodings follow a standard sinusoidal design. The query embedding $q \in \mathbb{R}^d$ remains unchanged. We concatenate the query with the enriched passage embeddings along the sequence length axis to form an input sequence, as shown in Figure 2, which is then processed by a stack of $M$ modified Transformer encoder layers.

Each Transformer encoder layer employs a hybrid attention mechanism composed of two modules. The *shared full attention* module is a standard multi-head attention mechanism that allows the query and all passages to attend to one another freely, capturing global relevance and inter-document relationships. In contrast, the *dedicated masked attention* module restricts each passage's attention to only those passages within the same document, plus the query, enabling fine-grained modeling of intra-document coherence. Formally, the attention score of this module is computed as:

$$\text{DedicatedAttnScore}(Q, K) = \text{softmax}\left(\frac{QK^\top + \texttt{mask}_{\text{doc}}}{\sqrt{d_k}}\right), \quad (1)$$

where $Q$ and $K$ are the query and key matrices, respectively, and $d_k$ is the head dimension. The additive attention mask $\texttt{mask}_{\text{doc}} \in \mathbb{R}^{(k+1)\times(k+1)}$ is defined such that the $(i, j)$-th entry is 0 if passage $j$ belongs to the same document as passage $i$ or if $j$ is the query embedding, and $-\infty$ otherwise. This masking encourages the model to reason locally within documents without interference from unrelated passages. As illustrated in Figure 2, this masking pattern ensures that: (i) the query can attend to all passages, (ii) passage $p_1$ from, e.g., document 0 can attend to the query and passage $p_2$ from the same document, but not to $p_3$ from, e.g., document $N$, and (iii) passage $p_3$ from document $N$ can attend only to the query and itself. Note that the dedicated attention mechanism always allows each passage to attend to the query, ensuring that the model can implicitly recognize the query position as special without any additional signals. To better illustrate how the two attention modules complement each other, consider the example below.

> **Query:** *When did Hull City win the 2016 Championship play-off final?*
> **Retrieved passages ($p_1$, $p_2$ are from Document A; $p_3$, $p_4$ are from Document B):**
> $p_1$: *Hull City is a professional football club based in Kingston upon Hull, England.*
> $p_2$: *They reached the 2016 Championship play-off final after finishing fourth in the league.*
> $p_3$: *The Championship play-off final determines which team is promoted to the Premier League.*
> $p_4$: *This match of 2016 took place on 28 May at Wembley Stadium.*

Within each document, the *dedicated masked attention* facilitates local reasoning, linking "*They*" to "*Hull City*" in Document A, and connecting "*This match*" to "*the Championship play-off final*" in Document B. Across documents, the *shared full attention* enables global interaction, aligning the "*Hull City*" in Document A with the event context in Document B to identify that passage $p_4$ contains the answer. Together, the two modules complement each other: the *dedicated masked attention* consolidates intra-document coherence, while the *shared full attention* integrates evidence across documents for accurate cross-passage reasoning. The outputs of the two modules are summed and passed through a feedforward network with residual connections and layer normalization. After $M$ stacked layers, we extract the updated passage embeddings $\{\hat{p}_1, \ldots, \hat{p}_k\}$ from the output sequence. These contextualized embeddings are then used to compute the final relevance scores.

**Training Objective.** Given a query $q$ and a set of candidate passages $\{x_1, \ldots, x_k\}$, each passage is labeled as either positive or negative based on its relevance to the query. Let $q \in \mathbb{R}^d$ denote

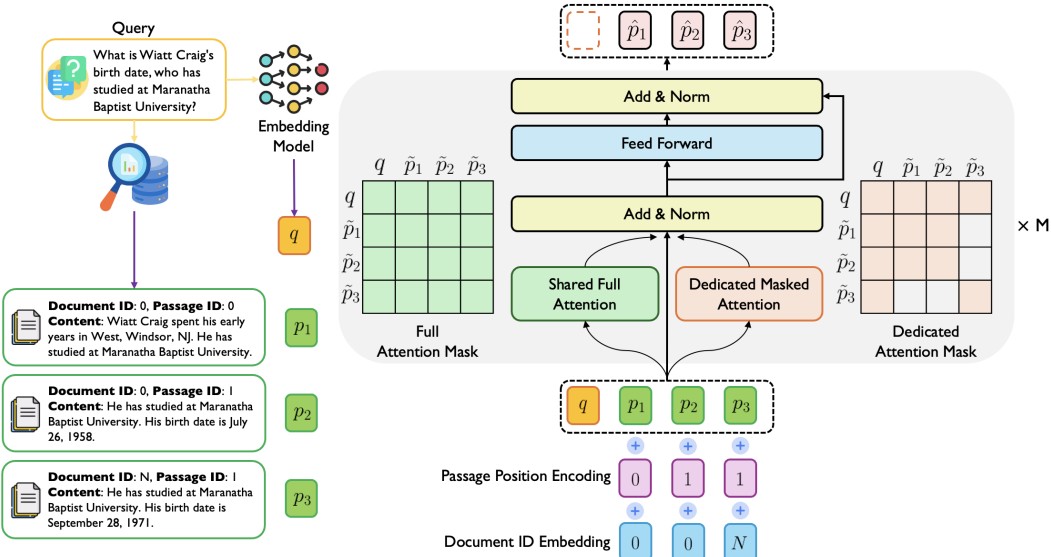

Figure 2: Overview of the EBCAR architecture. Candidate passage embeddings are enriched with document ID and passage position information, then processed jointly with the query embedding through a Transformer encoder. The hybrid attention mechanism combines a shared full attention module and a dedicated masked attention module. The outputs of the two modules are summed with residual connections and layer normalization.

the original query embedding, and $\{\hat{p}_1, \ldots, \hat{p}_k\}$, the updated passage embeddings obtained from the final Transformer encoder layer in EBCAR. Our training objective encourages the updated representation of the positive passage to be close to the query embedding, while pushing the negatives away. Therefore, we adopt a contrastive learning objective based on dot-product similarity and employ the InfoNCE loss (van den Oord et al., 2018) defined as:

$$\mathcal{L}_{\text{contrast}} = -\log \frac{\exp(\text{sim}(q, \hat{p}^+))}{\exp(\text{sim}(q, \hat{p}^+)) + \sum_j \exp(\text{sim}(q, \hat{p}_j^-))}, \tag{2}$$

where $\hat{p}^+$ is the updated embedding of the positive passage, $\{\hat{p}_j^-\}$ are the embeddings of the negative passages, and $\text{sim}(a, b) = a^\top b$ denotes dot-product similarity.

We intentionally use the unmodified query embedding $q$, instead of the query's updated embedding, to maintain a consistent reference point across candidate sets. This design choice avoids query-specific drift caused by the passage context and ensures that the passage representations are directly optimized to align with the static semantic anchor of the query. We validate this design in Appendix A.1, where we compare the training stability between fixed and updated query embeddings.

**Inference.** At inference time, we compute the relevance scores between the query and each candidate passage based on their final embeddings. Given the updated passage embeddings $\{\hat{p}_1, \ldots, \hat{p}_k\}$ and the static query embedding $q$, we compute $s_i = \text{sim}(q, \hat{p}_i)$, for $i = 1, \ldots, k$. The final passage ranking is obtained by sorting the scores $\{s_1, \ldots, s_k\}$ in descending order. This process is efficient and deterministic, leveraging the contextualized passage representations produced by EBCAR.

## 4 EXPERIMENTS

We conduct comprehensive experiments to evaluate the effectiveness and efficiency of our proposed EBCAR reranker. Our study is designed to answer the following research questions: **Q1**: Does EBCAR achieve competitive or superior performance compared to existing methods on datasets that require cross-passage inference? **Q2**: Is EBCAR more efficient at inference time than existing SOTA rerankers? **Q3**: Are EBCAR's architectural components, namely the passage position encoding and the hybrid attention mechanism, individually beneficial? **Q4**: How does EBCAR perform on datasets

that require less cross-passage inference, particularly in out-of-distribution settings? **Q5**: How does EBCAR perform when the number of retrieved passages scales up? To answer these questions, the remainder of this section is organized as follows: we first describe the datasets and evaluation protocol, followed by a summary of the baseline methods. We then present our main experimental results and provide an analysis of effectiveness and inference efficiency. Finally, we conduct ablation studies. In Appendix A.6 and Appendix A.7, we evaluate the generalization ability of EBCAR on datasets that do not emphasize cross-passage inference and the impact of retrieved candidate size.

## 4.1 DATASETS AND EVALUATION PROTOCOL

We conduct our evaluation on ConTEB[1] (Conti et al., 2025), a recently introduced benchmark designed to test whether retrieval models can leverage document-wide context when encoding individual passages. We use ConTEB as a challenging testbed for evaluating rerankers' abilities to perform context-aware reranking with cross-passage inference. Many of the benchmark datasets comprise long documents where relevant information is dispersed across chunks or coreferences and structural cues are essential for resolving ambiguity. Our experiments include both in-distribution evaluations, where test queries originate from the same domains as the training data, and out-of-distribution evaluations, where test queries are drawn from entirely different domains than training.

**Training Procedure.**   We train EBCAR on the ConTEB training splits: MLDR, SQuAD, and NarrativeQA. These datasets emphasize different forms of cross-passage inference, including entity-level disambiguation, fine-grained span matching, and narrative-level context comprehension. The training corpus consists of 311,337 passages and 362,142 training queries. For validation, we randomly sample 2,000 queries from the full set of 21,370 queries provided in the ConTEB validation split. To construct training examples, we use Contriever (Izacard et al., 2021) to retrieve the top-20 passages for each query. Since the gold passage is not guaranteed to appear among the retrieved candidates, we check its presence and, if absent, replace the $20^{\text{th}}$-ranked passage with the gold-positive one. This ensures that each training instance contains exactly one positive and up to nineteen negative passages. To avoid bias from retrieval rank, we randomly shuffle the passage order within each candidate set before feeding it to the model. It is worth mentioning that during inference and evaluation, we follow the same retrieval pipeline but do not enforce the inclusion of the gold passage; the top-20 candidates are taken directly from Contriever without any oracle intervention.

**Test Datasets.**   We evaluate EBCAR on eight datasets from the ConTEB benchmark, covering both in-distribution and out-of-distribution scenarios. **(i)** MLDR, **(ii)** SQuAD, and **(iii)** NarrativeQA (NarrQA) are used for both training and evaluation, enabling assessment of in-distribution performance for EBCAR and baselines trained on these three datasets. MLDR and NarrativeQA consist of encyclopedic and literary documents, respectively, where relevant information may be scattered across long passages. **(iv)** COVID-QA (COVID) is a biomedical QA dataset based on scientific research documents, which similarly requires reasoning over long and dense texts. **(v)** ESG Reports are long-form corporate documents sourced from the ViDoRe benchmark, re-annotated for text-based QA to test document-wide contextual understanding capabilities. **(vi)** Football and **(vii)** Geography (Geog) are constructed from rephrased Wikipedia summaries in which explicit entity mentions have been replaced by pronouns via GPT-4o. This setup introduces referential ambiguity and demands cross-passage entity disambiguation and coreference resolution to answer correctly. **(viii)** Insurance (Insur) comprises statistical reports on European Union countries, where country names often appear only in section headers, and therefore requires knowledge of a passage's structural position within the document to disambiguate country-specific information.

**Evaluation Metrics.**   We report nDCG@10 as our main effectiveness metric in Section 4.4, with MRR@10 results provided in Appendix A.2. Both metrics evaluate ranking quality and precision. We additionally measure inference efficiency using throughput, defined as the number of queries processed per second on a single A100 GPU with batch size 1. This dual-axis evaluation allows us to assess each reranker's practicality in latency-sensitive deployment settings.

---

[1]https://huggingface.co/collections/illuin-conteb/conteb-evaluation-datasets

Table 1: nDCG@10 across datasets. Orange denotes in-distribution tests, and Green denotes out-of-distribution tests. Throughput is the number of queries processed per second on a single A100 GPU. The best and second best results are **bolded** and underlined, respectively.

| Method | Training | Size | MLDR | SQuAD | NarrQA | COVID | ESG | Football | Geog | Insur | Avg | Throughput |
|---|---|---|---|---|---|---|---|---|---|---|---|---|
| BM25 | - | - | 65.42 | 45.07 | 36.20 | 39.79 | 10.10 | 4.69 | 23.66 | 0.00 | 28.12 | 263.16 |
| Contriever | - | - | 60.23 | 54.63 | 66.97 | 31.02 | 15.69 | 5.95 | 46.39 | 2.75 | 35.45 | 29.67 |
| monoBERT (base) | ConTEB Train | 110$M$ | 40.44 | 27.94 | 36.63 | 20.52 | 11.67 | 4.01 | 29.23 | 2.92 | 21.67 | 4.24 |
| monoT5 (base) | ConTEB Train | 223$M$ | 26.76 | 19.75 | 41.21 | 24.56 | 10.37 | 5.69 | 36.70 | 4.40 | 21.18 | 1.69 |
| monoT5 (base) | MS-MARCO | 223$M$ | 70.47 | 67.00 | 58.89 | 41.51 | 18.61 | 9.68 | 54.47 | 2.20 | 40.37 | 3.61 |
| duoT5 (base) | MS-MARCO | 223$M$ | 42.34 | 68.63 | 21.46 | 18.19 | 12.69 | 8.84 | 47.18 | 2.57 | 27.74 | 0.18 |
| RankVicuna | MS-MARCO | 7$B$ | 79.30 | 66.59 | 79.48 | 51.87 | 22.97 | 11.46 | 71.08 | 4.27 | 48.38 | 0.17 |
| RankZephyr | MS-MARCO | 7$B$ | 82.34 | 69.06 | **81.18** | 53.15 | 26.42 | 11.63 | 72.91 | 3.51 | 50.03 | 0.17 |
| FIRST | MS-MARCO | 7$B$ | 74.78 | 61.55 | 78.62 | 41.17 | 20.06 | 7.88 | 60.92 | 3.21 | 43.52 | 1.73 |
| FirstMistral | MS-MARCO | 7$B$ | 74.41 | 62.19 | 78.69 | 41.35 | 20.26 | 7.78 | 61.79 | 3.40 | 43.73 | 1.78 |
| LiT5Distil | MS-MARCO | 248$M$ | 60.33 | 54.87 | 66.94 | 31.22 | 16.07 | 5.98 | 46.66 | 2.78 | 35.61 | 0.66 |
| PE-Rank | MS-MARCO | 8$B$ | 52.58 | 45.00 | 51.58 | 33.40 | 19.92 | 6.29 | 46.25 | 1.37 | 32.05 | 0.81 |
| RankGPT (Mistral) | - | 7$B$ | 65.75 | 56.61 | 68.29 | 33.23 | 15.04 | 7.67 | 51.75 | 3.05 | 37.67 | 0.12 |
| RankGPT (Llama) | - | 8$B$ | 76.80 | 61.16 | 74.74 | 47.80 | 20.49 | 11.10 | 71.37 | 4.76 | 46.03 | 0.13 |
| ICR (Mistral) | - | 7$B$ | 79.49 | 67.41 | 79.50 | 52.54 | 25.65 | 10.57 | 71.44 | 4.00 | 48.83 | 0.18 |
| ICR (Llama) | - | 8$B$ | **83.93** | 69.09 | 79.96 | 53.32 | 28.36 | 10.91 | 73.10 | 4.16 | 50.35 | 0.19 |
| **EBCAR (ours)** | ConTEB Train | 126$M$ | 75.26 | **71.62** | 73.21 | **59.80** | **37.20** | **80.19** | **81.30** | **40.74** | **64.92** | 29.33 |

## 4.2 COMPARISON METHODS

We compare EBCAR against baselines spanning four categories: retriever-only, pointwise, pairwise, and listwise rerankers. All methods rerank the same set of 20 candidate passages retrieved by Contriever, without any oracle modification. *(a) Retriever-only baselines:* **(i)** BM25 is a classical sparse retriever based on lexical matching. **(ii)** Contriever (Izacard et al., 2021) is a dense retriever trained using unsupervised contrastive learning. *(b) Pointwise rerankers:* **(iii)** monoBERT (Nogueira et al., 2019) independently scores each query–passage pair using a binary classification head. **(iv)** monoT5 (Nogueira et al., 2020) casts reranking as a text generation task by producing the token "True" or "False." **(v)** monoT5 (MS-MARCO) refers to the same model trained on the MS-MARCO (Bajaj et al., 2016) dataset. *(c) Pairwise rerankers:* **(vi)** duoT5 (Pradeep et al., 2021) compares passage pairs to estimate their relative preference with respect to the query. *(d) Listwise rerankers:* **(vii)** RankVicuna (Pradeep et al., 2023a), **(viii)** RankZephyr (Pradeep et al., 2023b), and **(ix)** LiT5Distill (Tamber et al., 2023) are distilled listwise rerankers supervised by ranked outputs from GPT-style models. **(x)** FIRST (Gangi Reddy et al., 2024) and **(xi)** FirstMistral (Chen et al., 2024) use the logits of the first generated token to derive passage-level ranking scores. **(xii)** PE-Rank (Liu et al., 2025) encodes each passage into a single embedding for efficient listwise ranking. **(xiii)** RankGPT (Sun et al., 2023) prompts a language model to return a ranked list. **(xiv)** ICR (Chen et al., 2025) computes passage relevance using attention-based signals from in-context calibrated prompts over LLM representations. Due to limitations of computational resources, we only train monoBERT and monoT5 on ConTEB and evaluate other baselines with their provided checkpoints.

## 4.3 IMPLEMENTATION DETAILS

For training our EBCAR, we use the Adam optimizer (Kingma & Ba, 2015) with a fixed learning rate of $1 \times 10^{-3}$ and no weight decay. The model is trained for up to 20 epochs, with early stopping based on validation loss using a patience of 5 epochs. We set the batch size to 256. The default EBCAR configuration uses 16 layers, i.e., $M = 16$, a hidden size of 768, 8 attention heads. We employ these configurations to obtain similar amount of parameters as BERT base model. The implementation details of baseline methods are discussed in Appendix A.3. All experiments are run on a single A100 GPU.

## 4.4 RESULTS AND ANALYSIS

**Effectiveness.** We first compare the overall ranking quality of EBCAR with a broad set of baselines. As shown in Table 1, EBCAR achieves the highest average nDCG@10 across all test sets, outperforming all baselines. Our method shows particularly strong gains on tasks that require fine-grained entity disambiguation and coreference resolution, such as Football and Geography, where

reasoning across multiple passages and resolving entity mentions is essential. EBCAR also excels on Insurance, a task that heavily relies on positional information and structured document layouts; in this case, our architecture's ability to incorporate explicit passage position embeddings proves beneficial. These targeted improvements reflect the strengths of EBCAR in handling tasks that benefit from structured passage representation and alignment. We illustrate one sample from the football dataset and one from the insurance dataset in Appendix A.4 to highlight the underlying challenges.

On the remaining datasets, including both in-distribution datasets like MLDR, SQuAD, and NarrativeQA, and out-of-distribution datasets such as ESG and COVID, EBCAR maintains competitive performance. While it does not always surpass the most powerful LLM-based rerankers, e.g., ICR, we hypothesize that this is partly due to EBCAR's embedding-based architecture: since each passage is compressed into a fixed-size embedding, some fine-grained information may be lost, a limitation not shared by LLM rerankers that directly process the full text of each passage.

Despite this potential information bottleneck, EBCAR achieves strong and stable performance across domains. As shown in Figure 3(a), our model achieves the lowest average gap to the best method across all tasks, with narrow interquartile ranges, indicating stable performance. These results demonstrate that embedding-based approaches, when combined with architectural enhancements, can offer a robust and efficient alternative to heavyweight LLM-based rerankers.

**Efficiency.** We also evaluate the inference speed of each method, measured as the number of queries processed per second on a single A100 GPU. As shown in Table 1, EBCAR achieves a throughput of 29.33 queries per second, substantially outperforming all reranking baselines in terms of efficiency, while maintaining strong ranking quality. While models such as monoBERT and monoT5 have comparable parameter sizes to EBCAR, they process each query-passage pair independently, requiring $k$ separate forward passes for $k$ candidate passages per query. This leads to slower inference, especially in the case of pairwise models like duoT5, which compare passages in pairs and further increase the computational burden. At the other extreme, large LLM-based rerankers such as ICR or RankGPT are significantly slower due to both their large model sizes and their need to process the full textual content of all candidate passages. Even when some models like FIRST and FirstMistral do not explicitly generate output sequences, e.g., listwise orders like `[3] > [1] > [2]`, their long input contexts still cause substantial latency.

In contrast, EBCAR operates entirely in the embedding space, allowing all candidate passages to be encoded once independently. The final selection is performed through lightweight alignment computations, without the need for autoregressive generation or full-sequence processing. This design yields an order-of-magnitude speedup while maintaining strong performance, clearly positioning EBCAR on the Pareto frontier of ranking quality versus speed, as obviously shown in Figure 3(b).

**Summary.** Together, these results highlight that EBCAR offers a highly favorable integration of competitive ranking quality as well as fast and scalable inference.

Interestingly, we observe that monoBERT and monoT5 trained on ConTEB perform worse than Contriever or when trained on MS-MARCO. We hypothesize this stems from the fact that during training these pointwise models are forced to separate semantically similar passages from the gold passages while unable to utilize the document-wise context, which is the actual signal to distinguish the passages. This resulted in poor generalization. In contrast, EBCAR can leverage global context and structural signals across the candidate set, enabling more robust and context-aware ranking.

## 4.5 ABLATION STUDIES

We analyze the contribution of key components in EBCAR via ablation studies. Removing positional information (*w/o Pos*), which includes document ID embeddings and passage po-

Table 2: Ablation studies on two core components of EBCAR.

| Method | MLDR | SQuAD | NarrQA | COVID | ESG | Football | Geog | Insur |
|---|---|---|---|---|---|---|---|---|
| w/o Pos | 60.72 | 60.87 | 64.18 | 43.71 | 23.36 | 42.88 | 62.44 | 34.16 |
| w/o Hybrid | 74.55 | 47.52 | 66.32 | 42.93 | 34.44 | 41.93 | 60.34 | 36.00 |
| w/o Both | 43.46 | 40.13 | 51.80 | 29.63 | 15.77 | 5.28 | 43.70 | 2.88 |
| EBCAR | **75.26** | **71.62** | **73.21** | **59.80** | **37.20** | **80.19** | **81.30** | **40.74** |

sition encodings, leads to notable performance degradation, particularly on datasets like Insurance, where positional reasoning and document structure are critical. Ablating the hybrid attention mech-

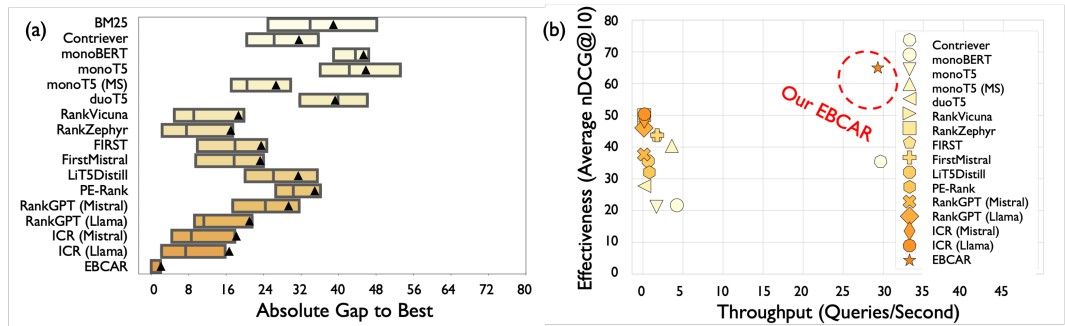

Figure 3: **(a)** For each method, we compute the gap in nDCG@10 from the best-performing method on each dataset. The boxes represent the interquartile range (25th to 75th percentile), vertical lines and black triangles indicate the median and mean, respectively. **(b)** An overall effectiveness-and-efficiency comparison across all methods. The effectiveness is measured by the average nDCG@10 across 8 test datasets. The efficiency is quantified by the throughput.

anism (*w/o Hybrid*) also results in performance drops on several datasets such as SQuAD. Interestingly, *w/o Hybrid* outperforms *w/o Pos* on datasets such as MLDR, NarrQA, and ESG, while the opposite trend is observed on SQuAD, COVID, and Football. These variations in direction reflect dataset-specific characteristics. Tasks that rely heavily on structural or positional cues (e.g., *Insurance*) are more affected by removing positional signals (w/o Pos), whereas datasets emphasizing semantic reconciliation across passages (e.g., *Football* and *Geography*) depend more on the hybrid attention for effective cross-passage reasoning (w/o Hybrid). This suggests that the two proposed components contribute complementarily to the overall performance of EBCAR. When both components are removed (*w/o Both*), performance deteriorates drastically across all datasets, underscoring the synergy between structural signals and the hybrid attention mechanism. These results validate the effectiveness of each component.

In Appendix A.5, we further analyze the hybrid attention mechanism through qualitative visualizations of its attention heatmaps. In Appendix A.6 and Appendix A.7, we evaluate EBCAR on datasets that do not emphasize cross-passage inference and the impact of retrieved candidate size.

## 4.6 ROBUSTNESS TO OTHER RETRIEVER

To examine the robustness of EBCAR to different embedding models, we replace the Contriever retriever and embeddings with E5 (Wang et al., 2024b), a stronger dense retriever. As shown by nDCG@10 scores, EBCAR maintains consistent performance improvements across datasets and further benefits from the stronger embedding space provided by E5. Specifically, the retriever-only E5 achieves 62.78 on MLDR, 55.86 on SQuAD, 67.12 on NarrQA, 33.74 on COVID, 15.38 on ESG, 6.05 on Football, 48.85 on Geography, and 4.02 on Insurance. When combined with EBCAR, the E5-based variant achieves 77.36 on MLDR, 72.68 on SQuAD, 75.04 on NarrQA, 61.07 on COVID, 36.88 on ESG, 80.64 on Football, 82.47 on Geography, and 41.24 on Insurance. These results confirm that EBCAR is retriever-agnostic and can seamlessly adapt to stronger embedding models while consistently improving overall retrieval quality.

## 5 CONCLUSION AND DISCUSSION

In this work, we introduce EBCAR, a novel embedding-based reranking framework designed to model context across multiple retrieved passages, by leveraging their structural signals and a hybrid attention mechanism. Extensive experiments on the ConTEB benchmark demonstrate that EBCAR outperforms strong baselines, especially on challenging datasets requiring entity disambiguation, co-reference resolution, and structural understanding.

## REPRODUCIBILITY STATEMENT

For reproducibility, we provide implementation details of our method in Section 4.3 and those of the baseline methods in Appendix A.3. The full codebase is released publicly at https://github.com/BorealisAI/EBCAR.

## LLM USAGE

We employed large language models exclusively to assist with polishing the writing of this paper. Specifically, LLMs were used to refine wording and verify grammar for clarity and readability. No aspects of method design, implementation, or experimental analysis involved LLM assistance.

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

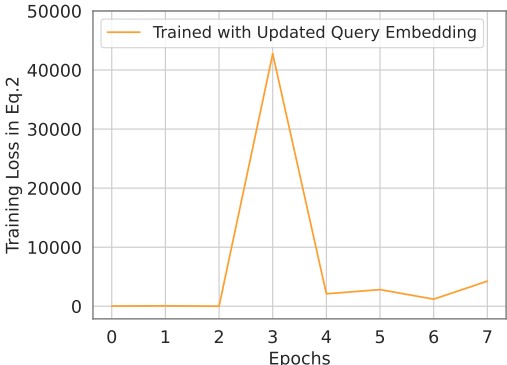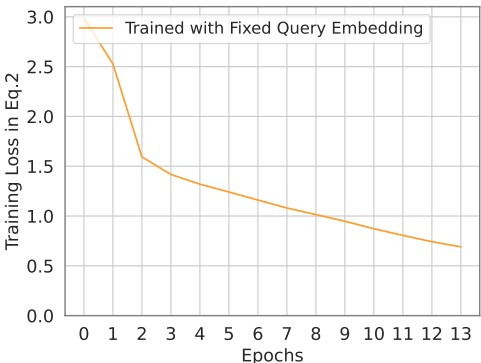

Figure 4: Training loss comparison between using an **updated query embedding** (left) and a **fixed query embedding** (right) in Eq. 2. Updating the query embedding leads to large and unstable loss spikes, while keeping it fixed produces smooth and monotonic convergence. This supports our design choice to use a fixed query embedding.

Table 3: MRR@10 across datasets. Orange denotes in-distribution tests, and Green denotes out-of-distribution tests. Throughput is the number of queries processed per second on a single A100 GPU. The best and second best results are **bolded** and underlined, respectively.

| Method | Training | Size | MLDR | SQuAD | NarrQA | COVID | ESG | Football | Geog | Insur | Avg | Throughput |
|---|---|---|---|---|---|---|---|---|---|---|---|---|
| BM25 | - | - | 61.35 | 42.90 | 36.18 | 37.35 | 8.56 | 3.72 | 23.46 | 0.00 | 26.69 | 263.16 |
| Contriever | - | - | 52.00 | 49.94 | 60.10 | 25.25 | 11.63 | 4.48 | 37.87 | 1.72 | 30.37 | 29.67 |
| monoBERT (base) | ConTEB Train | 110$M$ | 31.50 | 20.00 | 25.87 | 14.63 | 7.96 | 2.73 | 21.96 | 2.31 | 15.87 | 4.24 |
| monoT5 (base) | ConTEB Train | 223$M$ | 16.97 | 13.26 | 30.39 | 17.87 | 6.31 | 4.36 | 28.07 | 3.35 | 15.07 | 1.69 |
| monoT5 (base) | MS-MARCO | 223$M$ | 66.59 | 64.91 | 51.44 | 38.39 | 14.83 | 8.63 | 47.82 | 1.35 | 36.75 | 3.61 |
| duoT5 (base) | MS-MARCO | 223$M$ | 35.25 | 66.89 | 16.14 | 14.88 | 10.75 | 7.68 | 42.23 | 1.69 | 24.44 | 0.18 |
| RankVicuna | MS-MARCO | 7$B$ | 77.13 | 64.45 | 77.13 | 50.62 | 19.69 | 10.71 | 67.71 | 3.21 | 46.33 | 0.17 |
| RankZephyr | MS-MARCO | 7$B$ | 80.77 | 67.54 | **79.27** | 52.41 | 23.67 | 10.95 | 70.13 | 2.88 | 48.45 | 0.17 |
| FIRST | MS-MARCO | 7$B$ | 71.43 | 59.16 | 76.10 | 38.73 | 17.39 | 7.08 | 57.20 | 2.41 | 41.19 | 1.73 |
| FirstMistral | MS-MARCO | 7$B$ | 70.93 | 60.03 | 76.24 | 39.04 | 17.64 | 6.95 | 58.37 | 2.83 | 41.50 | 1.78 |
| LiT5Distil | MS-MARCO | 248$M$ | 52.10 | 50.18 | 60.10 | 25.46 | 12.01 | 4.53 | 38.15 | 1.75 | 30.54 | 0.66 |
| PE-Rank | MS-MARCO | 8$B$ | 41.32 | 35.93 | 39.34 | 25.42 | 16.04 | 4.48 | 35.34 | 0.52 | 24.80 | 0.81 |
| RankGPT (Mistral) | - | 7$B$ | 58.93 | 52.20 | 61.97 | 27.80 | 10.07 | 6.44 | 44.34 | 1.81 | 32.95 | 0.12 |
| RankGPT (Llama) | - | 8$B$ | 73.45 | 57.64 | 70.70 | 45.35 | 17.23 | 10.40 | 68.40 | 4.13 | 43.41 | 0.13 |
| ICR (Mistral) | - | 7$B$ | 77.04 | 65.61 | 78.63 | 52.53 | 23.21 | 9.76 | 69.27 | 2.85 | 47.36 | 0.18 |
| ICR (Llama) | - | 8$B$ | **82.93** | **67.83** | 79.18 | **53.64** | 25.38 | 10.18 | 71.33 | 2.87 | 49.17 | 0.19 |
| **EBCAR (ours)** | ConTEB Train | 126$M$ | 68.34 | 64.50 | 67.44 | 51.44 | **26.04** | **74.73** | **75.61** | **20.54** | **56.08** | 29.33 |

# A APPENDIX

## A.1 TRAINING WITH UPDATED QUERY EMBEDDING

To examine the effect of fixing the query embedding in the contrastive objective in Eq. 2, we compare two variants of EBCAR: one that uses the fixed input query embedding $q$, and another that uses the updated query representation produced by the encoder to compute the loss. As shown in Figure 4, training becomes highly unstable when the query embedding is updated. Loss values oscillate drastically and occasionally explode after a few epochs. In contrast, the fixed-query variant converges smoothly and yields stable optimization. We attribute this instability to *context drift*, where the query representation becomes entangled with passage-specific context, thus breaking its role as a consistent semantic anchor across candidate sets. These results empirically validate our design choice of using the unmodified query embedding in Eq. 2.

## A.2 SUPPLEMENTAL MRR@10 RESULTS

We provide full MRR@10 results in Table 3. The overall trends are consistent with our nDCG@10 results. EBCAR achieves the highest average score across all datasets. Notably, EBCAR excels on datasets such as Football, Geog, and Insurance, demonstrating its strong ability in handling tasks that

benefit from structured passage representation and alignment. These findings reinforce the motivations behind EBCAR and validate its robustness across varying cross-passage reasoning challenges.

### A.3 SUPPLEMENTAL IMPLEMENTATION DETAILS

For monoBERT and monoT5 trained on ConTEB, we implement these two models from scratch. For training, we use the Adam optimizer (Kingma & Ba, 2015) with a fixed learning rate of $2 \times 10^{-5}$ and $5 \times 10^{-5}$, respectively, as suggested in their original implementation details (Nogueira et al., 2019; 2020). These models are trained for up to 20 epochs, with early stopping based on validation loss using a patience of 5 epochs. We set the batch size to 32. All other baseline implementations are based on public code releases: monoT5 (MS-MARCO), duoT5, RankVicuna, RankZephyr, First-Mistral, and LiT5Distill are from the RankLLM library (Sharifymoghaddam et al., 2025)[2]. Following ICR (Chen et al., 2025), we use open-source models Mistral (Jiang et al., 2023)[3] and Llama-3.1 (et al., 2024)[4] for the RankGPT method, instead of GPT3.5/4 due to cost constraints. FIRST[5], PE-Rank[6], RankGPT and ICR[7] follow their respective original implementations. All experiments are run on a single A100 GPU.

### A.4 EXAMPLES OF FOOTBALL AND INSURANCE

> **Example 1 (Football).**
> **Query:** *What was Andrew Henry Robertson's first goal for Hull City?*
> **Gold passage:** *Despite the departure of several other first-team players, he chose to remain at City. His inaugural goal for the club was scored on 3 November 2015 in an away match against Brentford, where he initiated the scoring in a 2–0 victory that placed Hull at the top of the Championship table on goal difference. He participated in the 2016 Championship play-off final against Sheffield Wednesday, which Hull won 1–0 to achieve promotion to the Premier League. However, the team lasted only one season in the top division before being relegated again.*

All pronouns in this dataset are anonymized, so identifying that "*he*" refers to "*Andrew Henry Robertson*" requires reasoning across multiple passages mentioning the player and the club. EBCAR's hybrid attention enables such cross-passage coreference resolution effectively.

> **Example 2 (Insurance).**
> **Query:** What is the amount of (Re)insurance GWP in million in Austria?
> **Gold passage:** {*'General data': '— Amounts — Share total EEA —*
> *— Population (in 1000) — 8,979 — 2.00% —*
> *— (Re)insurance GWP (in million) — 20,815.873 — 1.5% —*
> *— Number of (re)insurance undertakings — 33 — 1.9% —*
> *— Number of registered insurance intermediaries — 17,999 — 2.1% —'*}

Without positional or structural cues, it is difficult to associate this numerical information with the correct country (Austria), which only appears in the section title. EBCAR's positional encodings provide this structural grounding, allowing accurate passage ranking.

### A.5 HYBRID ATTENTION ANALYSIS

To examine how the hybrid attention mechanism routes information, we conduct a qualitative study across all datasets. For each dataset, we randomly sample 20 test queries and, for every example, render an aggregated attention heatmap averaged over heads. We focus our analysis on the **final**

---

[2]https://github.com/castorini/rank_llm
[3]https://huggingface.co/mistralai/Mistral-7B-Instruct-v0.2
[4]https://huggingface.co/meta-llama/Llama-3.1-8B-Instruct
[5]https://github.com/gangiswag/llm-reranker
[6]https://github.com/liuqi6777/pe_rank
[7]https://github.com/OSU-NLP-Group/In-Context-Reranking

Table 4: nDCG@10 of selected baselines and EBCAR on TREC21, TREC22.

| Method | Training | Size | TREC2021 | TREC2022 | Average | Throughput |
|---|---|---|---|---|---|---|
| BM25 | - | - | 54.10 | 48.66 | 51.38 | 263.16 |
| Contriever | - | - | 64.19 | 61.36 | 62.78 | 29.67 |
| monoT5 | ConTEB Train | $223M$ | 52.57 | 46.66 | 49.62 | 1.69 |
| monoT5 | MS-MARCO | $223M$ | 85.04 | 76.92 | 80.98 | 3.61 |
| RankZephyr | MS-MARCO | $7B$ | 88.05 | 79.87 | 83.96 | 0.17 |
| ICR (Llama) | - | $8B$ | 75.28 | 67.47 | 71.38 | 0.19 |
| **EBCAR (ours)** | ConTEB Train | $126M$ | 72.77 | 66.65 | 69.71 | 29.33 |

**layer (Layer 15)** where the model's decision is consolidated. Figure 6 shows the **shared full attention** at Layer 15. Although weights vary across cells, attention is broadly distributed across passages, indicating global context exchange among all candidates. In contrast, Figure 7 shows the **dedicated masked attention** at Layer 15: information flow is sharply concentrated within passages that originate from the same document (block-diagonal patterns) and between the query and those passages. *While the figures present one representative case, we observe the same qualitative pattern consistently across the 20 sampled examples in every dataset*: the shared full attention supports global cross-passage reconciliation, whereas the dedicated masked attention strengthens intra-document consolidation. This matches EBCAR's design intent and explains the complementary gains seen in the ablations.

## A.6 PERFORMANCE ON MS-MARCO V2

To further evaluate the generalization ability of EBCAR, we compare it against selected rerankers on the MS-MARCO v2 dataset, using test queries from TREC 2021 and 2022. We select ICR (Llama) because it is the most recent and best-performing baseline on the ConTEB Test set, and thus serves as a strong comparator in this evaluation. RankZephyr and monoT5 are selected because they are the best-performing methods among LLM-supervised distillation methods and point-wise methods, respectively.

Compared to MS-MARCO v1, version 2 provides additional structural annotations such as document IDs and passage spans, making it a suitable testbed for evaluating EBCAR's ability to leverage structural signals (Bajaj et al., 2016). We utilize the ir_datasets library (MacAvaney et al., 2021) to access the judged queries from the TREC Deep Learning Tracks (2021 and 2022).

As shown in Table 4, EBCAR achieves comparable ranking performance to ICR (Llama), despite being significantly smaller in model size. Moreover, EBCAR exhibits a dramatically higher throughput, highlighting its practical advantage in scenarios that demand both efficiency and effectiveness.

While EBCAR performs competitively against ICR, it does not surpass monoT5 (MS-MARCO) or RankZephyr, both of which are pretrained and fine-tuned directly on MS-MARCO v1. We hypothesize two main reasons for this gap: (1) MS-MARCO v2 contains multiple golden passages per query. However, EBCAR is trained with the setting where there are only one golden passage among all candidate passages. This mismatch makes it harder for EBCAR to adapt to multi-relevant retrieval settings. (2) Both monoT5 (MS-MARCO) and RankZephyr are trained specifically on the MS-MARCO v1 dataset, which has overlaps with MS-MARCO v2 queries and documents, giving them an implicit domain advantage.

## A.7 IMPACT OF RETRIEVED CANDIDATE SIZE

In this section, we study how the performance of EBCAR varies as the number of retrieved candidate passages increases during inference. By default, EBCAR is trained with $k = 20$ passages, which defines the positional range of its document ID embeddings and sinusoidal position encodings. During inference, however, increasing $k$ beyond this value may result in extrapolation issues due to embedding mismatches or index overflows.

To study this issue, we train an additional variant of EBCAR using $k = 50$ passages and compare its performance against the default $k = 20$ variant, as well as against ICR (Llama), across varying numbers of retrieved passages from 10 to 50.

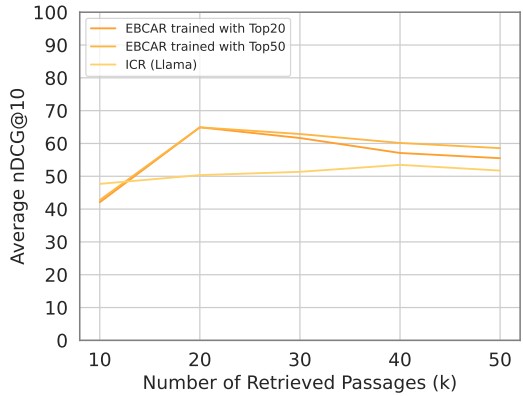

Figure 5: Impact of retrieved candidate size.

As shown in Figure 5, we observe that: **(i)** The default EBCAR (trained with Top20) degrades as $k$ increases after $k = 20$, likely due to extrapolation effects from unseen positional indices. **(ii)** The EBCAR trained with Top50 exhibits more stable performance across larger $k$ values and consistently outperforms EBCAR trained with Top20 variant beyond $k = 30$, validating our hypothesis about training-inference mismatch. **(iii)** Interestingly, both EBCAR variants exhibit a slight performance drop at large $k$, which aligns with prior findings (Jacob et al., 2024), where scaling up candidate size introduces noisy or distractive information that hampers reranking. **(iv)** Compared to ICR (Llama), both EBCAR variants outperform it across most $k$ values. These results suggest that it is better to train the EBCAR model with larger size of candidate passages during the training phase to mitigate the extrapolation issue for inference.

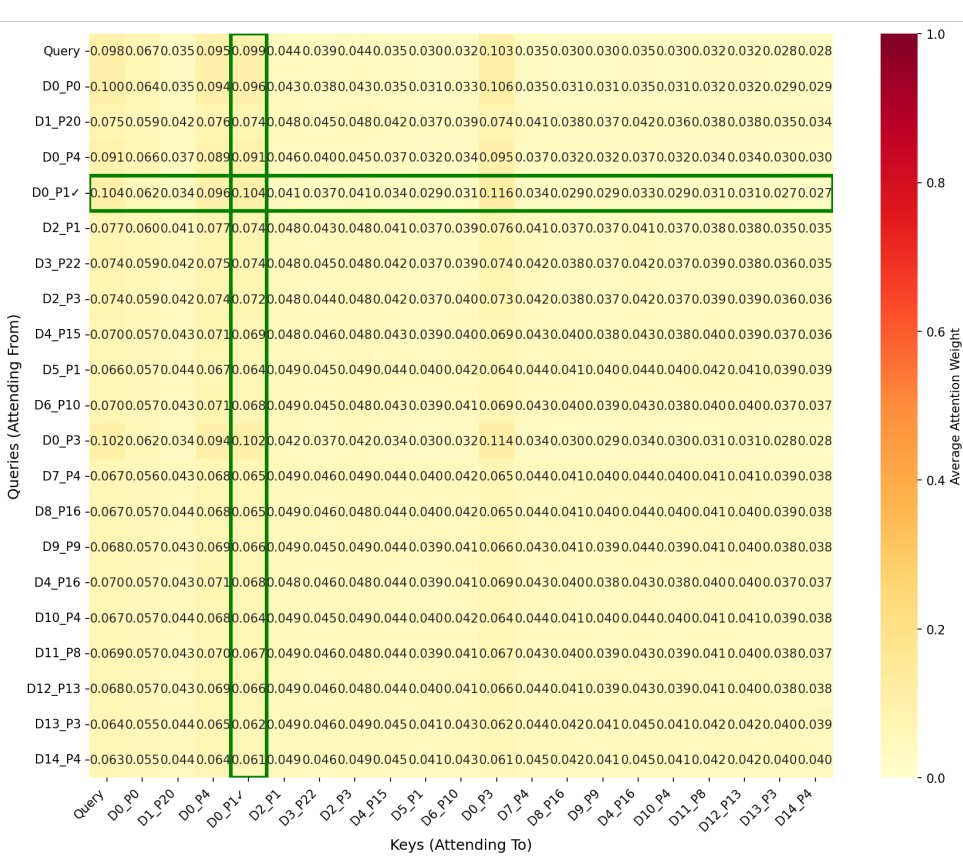

Figure 6: **Shared full attention** at **Layer 15**, averaged over all heads. The green box highlights the gold passage. Attention is more broadly distributed across passages, indicating global context exchange among all candidates.

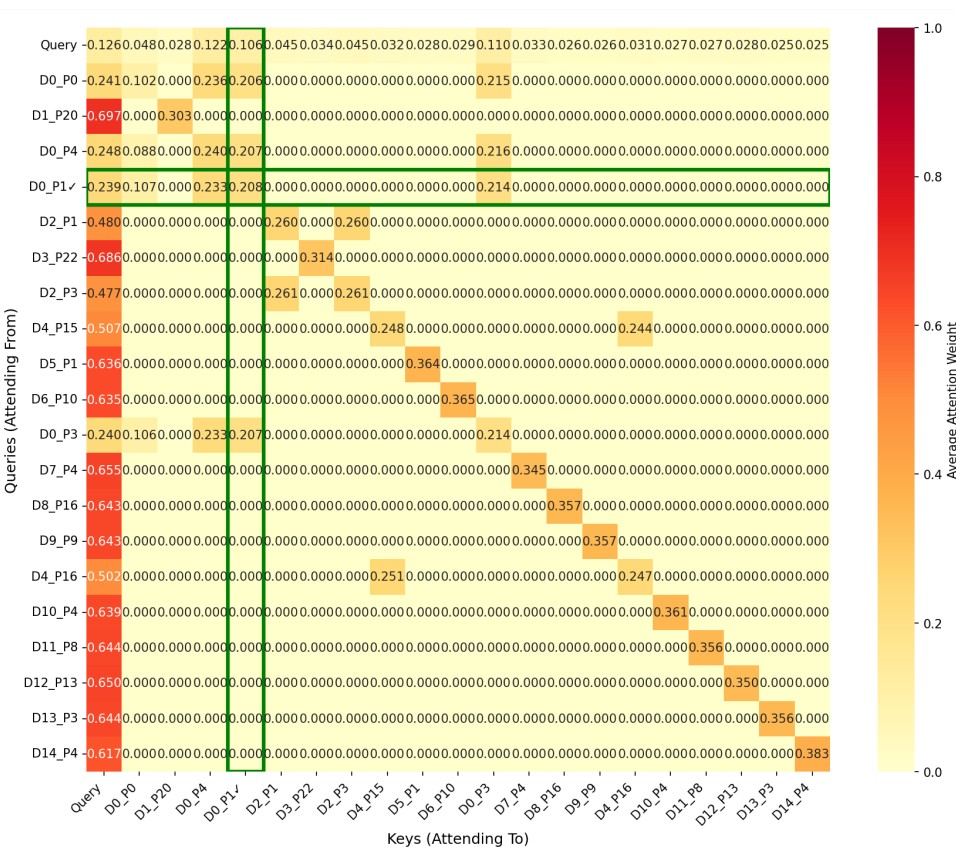

Figure 7: **Dedicated masked attention** at **Layer 15**, averaged over all heads. The green box highlights the gold passage. Attention concentrates within passages from the same document (block-diagonal structure) and between the query and those passages, reflecting document-aware locality.

