# OpenReview forum: "Embedding-Based Context-Aware Reranker"
_ICLR.cc/2026/Conference — ICLR 2026 Poster_

### Official Review · Reviewer_AjAm · 2025-10-29

**Soundness:** 2
**Presentation:** 3
**Contribution:** 2
**Rating:** 2
**Confidence:** 4

**Summary:**

The paper proposes EBCAR (Embedding-Based Context-Aware Reranker), a lightweight reranking framework that operates entirely in the embedding space. Experiments on the ConTEB benchmark demonstrate that EBCAR achieves strong performance in tasks requiring cross-passage entity disambiguation, coreference reasoning, and structural dependency modeling, while maintaining a favorable trade-off between accuracy and efficiency.

**Strengths:**

1. The method operates directly in the embedding space, avoiding repeated LLM-based text processing and improving inference speed.
2. The experiments are conducted on the ConTEB benchmark, which emphasizes cross-passage reasoning, and include both in-distribution and out-of-distribution evaluations, with clear and comprehensive results.

**Weaknesses:**

1. Since embeddings represent compressed information, fine-grained textual details may be lost. The paper does not conduct a quantitative analysis of which types of information are most susceptible to loss.
2. The model keeps the query embedding unchanged, yet no comparison is provided with an alternative setting where the query representation is updated. An analysis could clarify whether contextualizing the query improves or destabilizes semantic interactions.
3. The paper would benefit from qualitative analyses. Specifically, visualizing hybrid attention heatmaps for typical success and failure cases to illustrate how the model routes information within and across documents, thereby improving interpretability.
4. Most SOTA rerankers (e.g., RankVicuna, RankZephyr) are fine-tuned on MS-MARCO rather than ConTEB, whereas EBCAR is trained specifically on ConTEB, raising potential concerns about experimental fairness and comparability.
5. The terminology is inconsistent: document-aware masked attention (line 238) and dedicated masked attention (line 94) are used interchangeably and should be unified for clarity.

**Questions:**

1. Although the ablation study includes w/o Pos, w/o Hybrid, and w/o Both variants, the paper does not explain why the effects of these components differ in direction across certain datasets.

---

> ### Author Response · Authors · 2025-11-20
>
> We sincerely thank the reviewer for the thoughtful feedback and constructive suggestions. We appreciate the time and effort dedicated to carefully reviewing our paper. In the revised version, we have addressed each concern in detail by adding clarifications, new analyses, and additional experiments. All corresponding updates are highlighted in red throughout the paper for clarity. Below, we provide point-by-point responses to each of the reviewer’s comments.
>
> ## Weakness
>
> > Since embeddings represent compressed information, fine-grained textual details may be lost...
>
> We thank the reviewer for the insightful comment. Understanding which fine-grained information may be lost during embedding compression is indeed an interesting and important research question. However, this topic requires a systematic investigation of embedding representations and task-specific semantics, which is beyond the scope of our current work on developing an efficient cross-passage reranker. Prior studies have recognized the inherent information bottleneck in dense representations and explored ways to mitigate it [1–3]. Other work further proposes multi-view representation learning to capture complementary semantic aspects [4]. We therefore view this as a complementary research direction.
>
> [1] Wang et al., SimLM: Pre-training with Representation Bottleneck for Dense Passage Retrieval, ACL 2023.
>
> [2] Guo et al., Unsupervised Corpus-Aware Language Model Pre-training for Dense Passage Retrieval, ACL 2022.
>
> [3] Xiao et al., RetroMAE: Pre-Training Retrieval-Oriented Language Models via Masked Auto-Encoder, EMNLP 2022.
>
> [4] Zhang et al., Multi-View Document Representation Learning for Open-Domain Dense Retrieval, ACL 2022.
>
> > The model keeps the query embedding unchanged, yet no comparison is provided...
>
> We thank the reviewer for the comment. We intentionally compute the contrastive loss using the original query embedding $q$ rather than an updated query representation. This design maintains a consistent semantic reference and prevents drift caused by passage context. In this way, the passage representations are directly optimized to align with a fixed, stable query anchor, ensuring that the learned similarities reflect true query–passage relevance.
> In the revised version, we validate this design choice empirically. Specifically, we added a clarifying sentence in Sec.3 (Lines307–308): “We validate this design in AppendixA.1, where we compare the training stability between fixed and updated query embeddings.” AppendixA.1 presents this analysis: we compare two EBCAR variants, one using the fixed input query embedding and another using the updated query representation produced by the encoder to compute the loss. As shown in Figure4, updating the query embedding leads to unstable training, with oscillating and occasionally diverging losses, whereas the fixed-query variant converges smoothly and remains stable. We attribute this instability to *context drift*, where the query representation becomes entangled with passage-specific context, breaking its role as a consistent semantic anchor across candidate sets. These results empirically validate our design choice of using the unmodified query embedding in Eq.2.
>
> > The paper would benefit from qualitative analyses. Specifically, visualizing hybrid attention heatmaps...
>
> We thank the reviewer for the helpful suggestion. We have added qualitative visualizations of both the shared full attention and the dedicated masked attention scores in the revised manuscript. These heatmaps illustrate how the two attention modules route information within and across documents, improving interpretability.
> Specifically, we have included a qualitative analysis of the hybrid attention mechanism in Sec.4.5 (Lines509–510), which refers to a new detailed study in AppendixA.5 (Lines890–901). The results show that the shared full attention distributes weights globally across passages, while the dedicated masked attention concentrates within passages from the same document, confirming the expected complementary behaviors of the two modules and explaining the gains observed in the ablation study.

---

> ### Author Response · Authors · 2025-11-20
>
> > Most SOTA rerankers (e.g., RankVicuna, RankZephyr) are fine-tuned on MS-MARCO...
>
> We thank the reviewer for raising this point. Most test datasets in our evaluation are out-of-distribution with respect to the training set of ConTEB (as stated in Sec4.1). Even when excluding the three in-distribution datasets (MLDR, SQuAD, and NarrativeQA), EBCAR still achieves the highest nDCG@10 across all five out-of-distribution benchmarks (COVID, ESG, Football, Geography, and Insurance). In addition, we include in-distribution pointwise baselines (monoBERT and monoT5) trained on the same data to ensure a fair comparison.
>
> > The terminology is inconsistent: document-aware masked attention (line 238) and dedicated masked attention (line 94) are used interchangeably and should be unified for clarity.
>
> We thank the reviewer for pointing this out. We have unified the terminology to *dedicated masked attention* throughout the paper and update the figure caption accordingly.
>
> ## Questions
> > Although the ablation study includes w/o Pos, w/o Hybrid, and w/o Both variants, the paper does not explain why the effects of these components differ in direction across certain datasets.
>
> We thank the reviewer for the observation. The variations in direction reflect dataset-specific characteristics. Tasks that rely heavily on structural or positional cues (e.g., *Insurance*) are more affected by removing positional signals (*w/o Pos*), whereas datasets emphasizing semantic reconciliation across passages (e.g., *Football* and *Geography*) depend more on the hybrid attention for effective cross-passage reasoning (*w/o Hybrid*). The full EBCAR model integrates both components, consistently improving performance across all settings.
> In the revised version, we have added this clarification in Sec.4.5 (Lines485–504) to provide an interpretation that explains these variations and highlights the complementary effects of positional and hybrid attention.

---

> ### Author Response · Authors · 2025-11-24
> **Looking Forward to Your Feedback**
>
> Dear Reviewer AjAm,
>
> We hope this message finds you well. We would like to kindly follow up regarding the rebuttal feedback for our submission. We sincerely appreciate the time and effort you have invested in reviewing our work and completely understand how busy this period can be.
>
> If there are any additional clarifications or materials that would help in your evaluation, we would be more than happy to provide them. We truly value your consideration and the constructive insights you have shared.
>
> Thank you once again for your time and support.
>
> With kind regards,
>
> Authors

---

> ### Author Response · Authors · 2025-11-28
> **Follow Up**
>
> Dear Reviewer AjAm,
>
> We hope you are doing well. As the discussion deadline is approaching, we would like to kindly follow up on our rebuttal. We greatly appreciate the time and effort you have devoted to reviewing our paper, and we would be happy to provide any additional clarification if needed.
>
> Thank you again for your consideration and feedback.
>
> With kind regards,
> Authors

---

### Official Review · Reviewer_zHMo · 2025-10-29

**Soundness:** 3
**Presentation:** 4
**Contribution:** 4
**Rating:** 8
**Confidence:** 5

**Summary:**

This paper presents EBCAR: a new reranker model that works directly on the retrieved embeddings of the passages.

EBCAR concatenates the query and retrieved passages into one sequence, then calculates refined contextual embeddings for these passages in order to score them.  The refined embeddings are calculated by a  modified transformer architecture: a new dedicated masked attention module is added, where the attention is computed within passages of the same documents only. This document-masked attention is added along with the standard full-attention before the add+norm layer in the transformer encoder.

The paper has a comprehensive ablation and evaluation sections with multiple datasets covering various domains, showing uplift against strong baselines, and nearly a 7x throughput increase over the fastest neural reranker baseline in the experiments.

**Strengths:**

- The paper is easy to read and the contribution is clear.

-  The dedicated masked attention module is a novel idea that seems to work well in capturing intra-document coherence and semantics.

- The EBCAR model shows strong performance despite a very small size (126M parameters), showing significant uplift compared to much larger models.

- The throughput of EBCAR is impressive, this might be in part due to its smaller size considered in the experiments, but also due to working on the embeddings directly.

- The ablation studies in the appendix are comprehensive and justify the choices and architecture changes.

**Weaknesses:**

- Having a document ID embedding makes adding new documents difficult since a new embedding will need to be learned for it. It would be interesting to see experiments where the document embedding is computed from the token or passage embeddings to make adding new documents easier.

- Furthermore, learning a specific embedding per document makes me concerned about potential overfitting to these documents. It would be interesting to further analyze the content of these embeddings to see what concepts they captured that isn't already included in the tokens.

- I think a scaling study is lacking for this paper and would make it a lot stronger. Since the model is performing well on smaller scale, it would be interesting to see how the performance changes as we scale the model from 126M to 8B and from one GPU to 8 GPUs or even more, it would also be useful to understand how much of the impressive throughput is due to the architecture and embedding caching vs. the small number of parameters.

- Even though the paper focuses mostly on LLM-based reranker, but including models like BERT in the baseline drove me to wonder about the performance of a multi-vector ranking and retrieval methods like ColBERT in the same setup. Having this datapoint would be excellent for a fuller picture of how this model performs, especially to understand the impact of the information-bottleneck mentioned in the paper, since the multi-vector methods do not suffer from such information-bottleneck.

**Questions:**

- Figure 2 is very helpful, but it does not clearly show how the two attention modules are combined. Is the add an norm applied to all three sets of embeddings (raw, with shared full attention, and masked attention)

---

> ### Author Response · Authors · 2025-11-20
>
> We sincerely thank the reviewer for the thoughtful feedback and constructive suggestions. We appreciate the time and effort dedicated to carefully reviewing our paper. Below, we provide point-by-point responses to each of the reviewer’s comments.
>
> ## Weakness
>
> > Having a document ID embedding makes adding new documents difficult...
>
> We thank the reviewer for the comment. The document-ID embeddings in EBCAR are *relative and local* to each retrieved candidate set rather than global identifiers. As described in Sec.3 (Lines210–214), the table has at most $k$ rows, one per unique document within the retrieved candidates, and is reused across all queries. These embeddings are defined in a *relative* manner to distinguish documents within each candidate set and are *frozen* (non-trainable), so adding new documents to the corpus does not require any retraining.
> In the revised version, we further clarified this mechanism in Sec.3 (Lines214–226) by adding a toy example illustrating how document IDs are dynamically assigned and reused across different queries.
> While dynamically generating document embeddings from token or passage representations is an interesting direction, our current approach already supports dynamic document collections with negligible overhead. We plan to explore the reviewer’s suggested idea of generation-based document representations in future work.
>
> > Furthermore, learning a specific embedding per document makes me concerned about potential overfitting...
>
> We appreciate the reviewer’s concern. The "document-ID" embeddings in EBCAR are *relative and local* tags instantiated within each retrieved candidate set (up to $k$ rows) and reused across all queries. In our implementation, they are *frozen* (non-trainable), so no parameters are tied to specific documents. Their role is purely structural. They serve as content-agnostic indicators that identify which passages belong to the same source document, without encoding any semantic information beyond what passage embeddings already provide.
> To further prevent correlations with retrieval order, we randomly shuffle the passage order within each candidate set during both training and inference (Sec.4, Lines348–349). This design makes the relative document-ID assignment independent of retrieval rank and ensures permutation equivariance with respect to passage ordering, thereby preventing potential overfitting to retrieval patterns.
>
> > I think a scaling study is lacking for this paper and would make it a lot stronger...
>
> We thank the reviewer for this valuable suggestion. We fully agree that a scaling study would provide deeper insights into EBCAR’s efficiency and performance characteristics at larger model and hardware scales. As noted in Sec. 4.3 (Implementation Details), all our experiments were conducted on a single NVIDIA A100 GPU due to limited computational resources available to us. Consequently, we were unable to perform large-scale experiments (e.g., scaling from 126 M to 8 B parameters or across multiple GPUs). We share the reviewer’s interest in exploring EBCAR’s scalability, and we plan to include such experiments in future work when additional resources become available. We hope the reviewer understands this constraint.
>
> > Even though the paper focuses mostly on LLM-based reranker, but including models like BERT in the baseline drove me to wonder about the performance of a multi-vector ranking...
>
> We thank the reviewer for the thoughtful suggestion. Multi-vector methods such as ColBERT indeed somehow alleviate the single-vector "information bottleneck," but they come with much higher computational costs. In EBCAR, each passage is represented by a single embedding, resulting in a per-layer attention complexity of $\mathcal{O}(k^2d)$. If using a multi-vector retriever like ColBERT, each passage would contain tens of token-level embeddings, expanding the input length to roughly $k\times n$ (where $n$ is the number of vectors per passage) and increasing the complexity to $\mathcal{O}((kn)^2d')$ (where $d'$ is the dimension of ColBERT's embedding), which would significantly degrade efficiency and make EBCAR’s high throughput unattainable. To maintain comparable efficiency, one would need to pool or compress ColBERT’s vectors, which reintroduces the same information bottleneck. For these reasons, we focus on the single-vector setup in this work, while we acknowledge the reviewer’s interesting suggestion and plan to explore multi-vector extensions of EBCAR in future work.

---

> > ### Author Response · Authors · 2025-11-20
> >
> > ## Questions
> >
> > > Figure 2 is very helpful, but it does not clearly show how the two attention modules are combined. Is the add an norm applied to all three sets of embeddings (raw, with shared full attention, and masked attention)
> >
> > We appreciate the reviewer’s question and confirm that the interpretation is correct. In each Transformer layer, the outputs from the shared full attention and the dedicated masked attention modules are summed together, followed by a residual connection from the input ("raw") embeddings and then a LayerNorm operation. In other words, the layer output is computed as $\mathrm{LayerNorm}(x + \mathrm{FullAttn}(x) + \mathrm{MaskedAttn}(x))$, where $x$ denotes the input embeddings. In the revised version, we have explicitly clarified this combination in the caption of Figure2, adding the sentence: "The outputs of the two modules are summed with residual connections and layer normalization."

---

> ### Author Response · Authors · 2025-11-24
> **Appreciation for Your Feedback**
>
> Dear Reviewer zHMo,
>
> We sincerely thank you for your thoughtful and encouraging feedback on our submission. Your insights are greatly appreciated and have been very helpful in improving our work.
>
> With kind regards,
>
> Authors

---

### Official Review · Reviewer_dequ · 2025-10-30

**Soundness:** 3
**Presentation:** 3
**Contribution:** 3
**Rating:** 4
**Confidence:** 4

**Summary:**

The paper proposes Embedding-based Context-Aware Reranker (EBCAR), a reranking model designed to address the challenges of efficiency and cross-passage inference in passage reranking tasks. To improve efficiency, EBCAR operates directly on the embeddings of the query and passages, rather than their raw text representations. To enhance cross-passage reasoning, it augments each passage embedding with two additional components: a document ID embedding and a passage position embedding, enabling the model to better capture contextual and structural information across passages. Furthermore, the paper introduces a hybrid attention mechanism to refine the passage embeddings, which are then used to produce the final ranking.

**Strengths:**

1. The paper proposes a novel embedding-based context-aware reranker that improves both efficiency and cross-passage inference.
2. Experimental results on eight datasets show that the proposed approach achieves the best average score.
3. The paper is well-written and easy to follow.

**Weaknesses:**

1. EBCAR requires learning a fixed-size document ID embedding matrix of size k × d, which limits scalability. Expanding to a larger document set would require retraining the model to accommodate new document IDs, making it less flexible in dynamic or large-scale retrieval settings.
2. As shown in Table 1, the proposed model achieves exceptionally high performance on the Football and Insur datasets (e.g., 80.19 vs. 11.63, and 40.74 vs. 4.76 compared to the best baseline), which contributes significantly to the overall average improvement. However, the paper lacks insight into why the model performs so well on these datasets. Moreover, the lack of detailed descriptions of each dataset makes it difficult to understand and analyze the differences in model performance.
3. The reranking experiments are conducted only on passages retrieved by Contriever, making it unclear whether the proposed method can generalize to results retrieved by stronger retrievers such as E5 or BGE. Evaluating on diverse retrieval backbones would strengthen the validity and robustness of the proposed approach.
4. The paper lacks discussion and comparison with document-level retrievers or rerankers, which are relevant baselines given the cross-passage inference objective.

**Questions:**

1. How scalable is EBCAR to larger or dynamic document collections, given that the document ID embedding matrix has a fixed size?
2. Can the authors provide insights into why EBCAR achieves such large gains on the Football and Insur datasets? Are there characteristics of these datasets that particularly favor the proposed method?
3. Has the model been tested with retrieval results from stronger retrievers like E5 or BGE? If not, how confident are the authors that EBCAR would generalize to different retrieval backbones?

---

> ### Author Response · Authors · 2025-11-20
>
> We sincerely thank the reviewer for the thoughtful feedback and constructive suggestions. We appreciate the time and effort dedicated to carefully reviewing our paper. In the revised version, we have addressed each concern in detail by adding clarifications, new analyses, and additional experiments. All corresponding updates are highlighted in red throughout the paper for clarity. Below, we provide point-by-point responses to each of the reviewer’s comments.
>
> ## Weakness & Questions
> > EBCAR requires learning a fixed-size document ID embedding matrix...
>
> > How scalable is EBCAR to larger or dynamic document collections, given that the document ID embedding matrix has a fixed size?
>
> We thank the reviewer for the comment. The document-ID embeddings in EBCAR are *relative and local* to each retrieved candidate set rather than global identifiers. As described in Sec.3 (Lines210–214), the table has at most $k$ rows, one per unique document within the retrieved candidates, and is reused across all queries. These embeddings are defined in a *relative* manner to distinguish documents within each candidate set and are *frozen* (non-trainable), so adding new documents to the corpus does not require any retraining.
> In the revised version, we further clarified this mechanism in Sec.3 (Lines214–226) by adding a toy example illustrating how document IDs are dynamically assigned and reused across different queries. This example shows that the same small embedding table is re-indexed on the fly for each query, demonstrating that the mechanism scales naturally to large and dynamic corpora.
>
> > As shown in Table 1, the proposed model achieves exceptionally high performance on the Football and Insur
>
> > Can the authors provide insights into why EBCAR achieves such large gains on the Football and Insur datasets? Are there characteristics of these datasets that particularly favor the proposed method?
>
> We appreciate the reviewer’s insightful comment. The characteristics of these datasets and the reasons behind the observed performance differences are detailed in Sec.4.1 (Lines352–365) and Sec.4.4 (Lines425–433). Specifically, *Football* and *Geography* anonymize entity mentions by replacing them with pronouns, introducing referential ambiguity that necessitates cross-passage coreference resolution, whereas *Insurance* comprises structured statistical reports where country names appear only in section headers, requiring positional and structural reasoning. EBCAR demonstrates substantial gains on these datasets because the dedicated masked attention facilitates information exchange among passages from the same document (resolving entity references), while the positional encodings supply explicit structural cues that enhance within-document alignment. On the remaining datasets (e.g., MLDR, SQuAD, COVID, ESG), where structural or referential dependencies are weaker, the improvements are smaller yet remain competitive; this aligns with our discussion in Sec.4.4 (Lines435–439).
> In the revised version, we further clarified this by adding a sentence in Sec.4.4 (Lines432–433): "We illustrate one sample from the Football dataset and one from the Insurance dataset in AppendixA.4 to highlight the underlying challenges." AppendixA.4 (Lines888–914) now presents two representative examples from these datasets, illustrating the reasoning types required for cross-passage entity resolution and structural disambiguation. Additional examples can be found in the public ConTEB benchmark.

---

> ### Author Response · Authors · 2025-11-20
>
> > The reranking experiments are conducted only on passages retrieved by Contriever, making it unclear whether the proposed method can generalize...
>
> > Has the model been tested with retrieval results from stronger retrievers like E5 or BGE? If not, how confident are the authors that EBCAR would generalize to different retrieval backbones?
>
> We thank the reviewer for the valuable suggestion. EBCAR is retriever-agnostic, as it operates directly on dense passage and query embeddings regardless of their source. To further verify this robustness, we have added new experiments in Sec.4.6 (Lines514–523), where we replace the Contriever retriever and embeddings with E5, a stronger dense retriever.
> As shown by the nDCG@10 results, EBCAR maintains consistent performance improvements across datasets and further benefits from the stronger embedding space provided by E5. These findings confirm that EBCAR generalizes well across different retrieval backbones and can seamlessly adapt to stronger embedding models without any architectural modification.
>
> > The paper lacks discussion and comparison with document-level retrievers or rerankers...
>
> We thank the reviewer for the comment. Our work specifically focuses on the passage-level reranking problem, as the need for cross-passage inference primarily arises from the chunking process in RAG pipelines, where long documents are split into shorter passages for retrieval. In contrast, document-level retrievers and rerankers operate on entire documents and address a different problem formulation that does not involve passage-level segmentation. Therefore, these approaches are not directly comparable to our setting.
> As discussed in Sec.2 (Lines128–131), we explicitly position EBCAR as complementary to such document-level approaches. While document-level methods aim to improve retrieval granularity or representation at the document scale, EBCAR enhances cross-passage reasoning within the reranking stage, providing an additional layer of context integration on top of existing strategies.

---

> ### Author Response · Authors · 2025-11-24
> **Looking Forward to Your Feedback**
>
> Dear Reviewer dequ,
>
> We hope this message finds you well. We would like to kindly follow up regarding the rebuttal feedback for our submission. We sincerely appreciate the time and effort you have invested in reviewing our work and completely understand how busy this period can be.
>
> If there are any additional clarifications or materials that would help in your evaluation, we would be more than happy to provide them. We truly value your consideration and the constructive insights you have shared.
>
> Thank you once again for your time and support.
>
> With kind regards,
>
> Authors

---

> ### Author Response · Authors · 2025-11-28
> **Follow Up**
>
> Dear Reviewer dequ,
>
> We hope you are doing well. As the discussion deadline is approaching, we would like to kindly follow up on our rebuttal. We greatly appreciate the time and effort you have devoted to reviewing our paper, and we would be happy to provide any additional clarification if needed.
>
> Thank you again for your consideration and feedback.
>
> With kind regards,
> Authors

---

### Official Review · Reviewer_VaSb · 2025-11-02

**Soundness:** 2
**Presentation:** 3
**Contribution:** 2
**Rating:** 4
**Confidence:** 3

**Summary:**

EBCAR is a novel reranking method to solve the critical challenge of cross-passage inference, ignored by other methods. Specifically, it incorporates the structural information of passages via a hybrid attention mechanism to capture both high-level inter-document and low-level intra-document interactions. Combining evidences from multiple passages, the performance improvement on benchmark datasets shows its effectiveness and efficiency.

**Strengths:**

--Cross-passage inference, the core problem studied in this paper, is critical for complex reasoning tasks.

--NDCG@10 of the proposed method is consistently better than state-of-the-art baselines on benchmark datasets.

--It is more efficient compared with rerankers based on large and high-cost language models.

**Weaknesses:**

--Both the hybrid attention and the positional encoding are two key components for EBCAR according to the ablation study in Table 2. However, how the hybrid attention mechanism improves the cross-passage understanding is still a question to be explored and explained further. Maybe a detailed analysis of the hybrid attention should be provided and also an intuitive example is used to illustrate their relationship.

--Compared with the efficiency of Contriever, the advantage of the proposed method EBCAR is not obvious. Its time complexity needs to be analyzed for complementary.

--The performance improvement of the proposed method in Table 1 is 8~10 times of state-of-the-art baseline methods on Football and Insure, while the performance improvement is not so large on other datasets. The underlying reason should be clarified.

**Questions:**

-The proposed embedding based rerank method is highly dependent on the embedding model, such as BERT, Mistral and Llama. It is unfair to compare reranking models with different embedding models listed in both Table 1 and 3.

-The performance comparison between EBCAR and ICR in Table 1 under the evaluation of NDCG@10 is inconsistent with their comparison result in Table 3 under the evaluation of MRR@10. The underlying reasons are to be explored.

-How the reranked results’ influence on the generation results should be further explored.

---

> ### Author Response · Authors · 2025-11-20
>
> We sincerely thank the reviewer for the valuable feedback and constructive comments. We appreciate the time and effort devoted to carefully reviewing our paper. We have addressed each concern in detail below, providing clarifications, additional analyses, and new experiments added to the revised manuscript. All corresponding changes are highlighted in red in the updated version for clarity.
>
> ## Weakness
> > Both the hybrid attention and the positional encoding are...
>
> We thank the reviewer for the helpful suggestion. Concretely, our hybrid attention consists of two parallel modules: the *shared full attention* models global interactions among the query and all passages (capturing inter-document relationships), while the *dedicated masked attention* focuses on local interactions within the same document and the query (ensuring intra-document coherence). Their outputs are combined via residual connections and layer normalization.
> In the revised version, we have added an intuitive example in Sec.3 (Lines248–263) to illustrate how the two attention modules complement each other. The example demonstrates a query with four retrieved candidate passages involving both intra-document reasoning (coreference resolution) and inter-document alignment, showing how the shared and dedicated attentions cooperate to identify the correct evidence.
> In addition, we have included a qualitative analysis of the hybrid attention mechanism in Sec.4.5 (Lines509–510), which refers to a new detailed study in AppendixA.5 (Lines890–901). The results show that the shared full attention distributes weights globally across passages, while the dedicated masked attention concentrates within passages from the same document, confirming the expected complementary behaviors of the two modules and explaining the gains observed in the ablation study.
>
> > Compared with the efficiency of Contriever...
>
> Thank you for the comment. Contriever is a retriever-only baseline that encodes the query and each passage independently to enable coarse-grained similarity search in embedding space. In contrast, EBCAR is a reranker that jointly models interactions among the query and all retrieved passages to refine their relevance scores. Reranking is a typically more computationally demanding process. Despite this, EBCAR achieves near-retriever speed (29.33 vs. 29.67 queries/second) while substantially improving ranking accuracy. Theoretically, one EBCAR layer costs $\mathcal{O}(k^2 d)$ for attention and $\mathcal{O}(kd^2)$ for feed-forward computation, which remains lightweight since it operates on embedding-level representations rather than token sequences. In comparison, text-level rerankers such as RankGPT require $\mathcal{O}(L^2 d + L d^2)$ operations with $L{\approx}500k$ tokens when $k$ passages (assume $\approx 500$ tokens each) are concatenated.
>
> > The performance improvement of the proposed method in Table 1 is 8 to 10 times...
>
> We appreciate the reviewer’s observation. In Sec.4.1 (Lines352–365), we describe that *Football* and *Geography* replace explicit entity mentions with pronouns, introducing referential ambiguity that requires cross-passage coreference resolution, while *Insurance* consists of structured statistical reports where country names appear only in section headers, requiring positional and structural inference. In Sec.4.4 (Lines425–433), we further analyze that EBCAR achieves the largest gains precisely on these datasets because its dedicated masked attention allows passages from the same document to exchange information (resolving entity references), and its positional encodings provide explicit structural cues for within-document alignment. On the remaining datasets (e.g., MLDR, SQuAD, COVID, ESG), where structural or referential dependencies are weaker, the improvements are smaller but still competitive; this is consistent with our discussion in Sec.4.4 (Lines435–439).
> In the revised version, we have also added a clarifying sentence in Sec.4.4 (Lines432–433): "We illustrate one sample from the Football dataset and one from the Insurance dataset in AppendixA.4 to highlight the underlying challenges." AppendixA.4 (Lines888–914) now includes two representative examples from these datasets, demonstrating the types of reasoning required for cross-passage entity resolution and structural disambiguation. More examples can be found in the public ConTEB benchmark dataset.

---

> ### Author Response · Authors · 2025-11-20
>
> ## Questions
> > The proposed embedding based rerank method is highly dependent on the embedding model...
>
> We thank the reviewer for the comment. All rerankers operate on the *same* top-20 retrieved candidates, ensuring identical inputs for a fair comparison. EBCAR itself is embedding-function agnostic, which we further verify in Sec.4.6 (Lines514–523) through an additional experiment replacing Contriever with E5 as the embedding model. EBCAR can take any fixed-size passage and query embeddings as input, and in our main experiments we use Contriever embeddings for consistency. In contrast, text-based rerankers such as RankVicuna, RankZephyr, LiT5Distill, FIRST, RankGPT, and ICR process raw text rather than embeddings; their use of different backbone LLMs (e.g., Mistral, Llama, T5) reflects a fundamentally different reranking paradigm, not an inconsistency in the experimental setup.
>
> > The performance comparison between EBCAR and ICR in Table 1 under the evaluation of NDCG@10 is inconsistent...
>
> We thank the reviewer for the observation. The apparent inconsistency between Table1 (nDCG@10) and Table3 (MRR@10) arises from the inherent difference between the two metrics. MRR@10 focuses solely on the exact rank of the first relevant passage, while nDCG@10 applies a logarithmic discount and is less sensitive when the gold passage appears slightly below the top rank.
> For example, if the ranking of the gold passage returned by a model is $i$, its MRR is $\frac{1}{i}$, but its nDCG is $\frac{1}{\log_2(i+1)}$. Assume model A and B ranked 14 examples with rankings 1 (3 examples), 3 (1 example), 7 (10 examples) for A and 3 (14 examples) for B. Then MRR would be (3/1 + 1/3 + 10/7)/14 = 0.34 for A and 0.33 for B, but nDCG would be (3/1 + 1/2 + 10/3)/14 = 0.49 for A and 0.50 for B.
> Therefore, different relative orderings on a few datasets (e.g., SQuAD and COVID) are expected and consistent with the metrics’ definitions. As in most prior reranking studies, we adopt nDCG@10 as the primary effectiveness metric and report MRR@10 as an additional reference.
>
> > How the reranked results’ influence on the generation results should be further explored.
>
> We thank the reviewer for the suggestion. While rerankers are often used within RAG pipelines, our work focuses on the reranking model itself, specifically, improving its ability to order retrieved passages by relevance. Following prior reranking studies [1, 2], we evaluate using standard ranking metrics (nDCG@10 and MRR@10), which are well-established measures of reranker effectiveness. A full evaluation of end-to-end generation performance is beyond the scope of this work, and we therefore omit such experiments.
>
> [1] Sun et al., Is ChatGPT Good at Search? Investigating Large Language Models as Re-Ranking Agents, EMNLP 2023.
>
> [2] Chen et al., Attention in Large Language Models Yields Efficient Zero-Shot Re-Rankers, ICLR 2025.

---

> ### Author Response · Authors · 2025-11-24
> **Looking Forward to Your Feedback**
>
> Dear Reviewer VaSb,
>
> We hope this message finds you well. We would like to kindly follow up regarding the rebuttal feedback for our submission. We sincerely appreciate the time and effort you have invested in reviewing our work and completely understand how busy this period can be.
>
> If there are any additional clarifications or materials that would help in your evaluation, we would be more than happy to provide them. We truly value your consideration and the constructive insights you have shared.
>
> Thank you once again for your time and support.
>
> With kind regards,
>
> Authors

---

> ### Author Response · Authors · 2025-11-28
> **Follow Up**
>
> Dear Reviewer VaSb,
>
> We hope you are doing well. As the discussion deadline is approaching, we would like to kindly follow up on our rebuttal. We greatly appreciate the time and effort you have devoted to reviewing our paper, and we would be happy to provide any additional clarification if needed.
>
> Thank you again for your consideration and feedback.
>
> With kind regards,
> Authors

---

### Author Response · Authors · 2025-12-03
**Rebuttal Summary**

We thank the AC for handling our submission and for coordinating the thoughtful reviews. Below we briefly summarize the consensus strengths and how the rebuttal addresses the reviewers’ key concerns.

**Reviewers agree that EBCAR is a novel, lightweight embedding-based reranker that explicitly targets cross-passage inference while maintaining strong efficiency**. They highlight the clarity of the paper, the hybrid attention design (shared full attention + dedicated masked attention) for inter-/intra-document reasoning, the comprehensive ablations, and the strong effectiveness/throughput trade-off.

Most concerns focused on clarifying *how* EBCAR works and *why* it performs especially well on certain datasets, as well as on fairness and scalability. In the revision, we:
- Added an intuitive example and qualitative attention heatmaps to show how hybrid attention cooperates for cross-passage reasoning, and provided dataset-level interpretations for ablation results.
- Explained in detail why Football/Insurance benefit most from EBCAR (pronoun-based entity ambiguity and structural/statistical layout), and added representative examples in the appendix.
- Provided a time-complexity analysis contrasting embedding-level attention with token-level LLM rerankers, emphasizing that EBCAR achieves near-retriever throughput while modeling cross-passage interactions.
- Clarified that document-ID embeddings are **local, relative, and frozen**, reassigned per query, so adding new documents requires no retraining and does not induce content overfitting.
- Added experiments with a stronger retriever (E5) to show that EBCAR is retriever-agnostic and continues to improve over the new backbone.
- Justified keeping the query embedding fixed via a new stability analysis, showing that updating the query leads to training instability due to context drift.
- Addressed fairness concerns by clarifying that all rerankers operate on the same retrieved candidates and that we include in-distribution pointwise baselines trained on the same data.

Some broader directions raised by reviewers (e.g., scaling to larger models/multi-GPU and integrating multi-vector retrievers like ColBERT) are acknowledged as important but are beyond our current compute budget and are left as future work. **Overall, we believe the revisions substantially strengthen the paper and resolve the main technical and empirical concerns.**

**We hope the AC will consider these revisions and the substantial clarifications provided when making the final decision. We sincerely appreciate the AC’s time, effort, and service to the community.**

---

### Meta-Review · Area_Chair_hMeg · 2026-01-05

**Summary:**

The paper proposes a method for re-ranking in settings requiring cross-passage inference. The key idea is to employ a hybrid attention scheme with a combination of conventional inter-document attention, and intra-document attention. This attention scheme is argued to provide a combination of local representation learning, and conventional inter-document relationships. The authors demonstrate strong results on the ConTEB benchmark.

Reviewers were generally appreciative of the paper's effective approach. However, they also raised a number of concerns:
- **Limitations of document ID embedding**. Multiple reviewers noted that by learning separate parameters for each document, there is a risk of overfitting, limited scalability, and limited generalization to new documents.
- **Comparison with document-level and multi-vector re-rankers**. Multiple reviewers noted that the importance of cross-passage interaction motivates a comparison against a document-level re-rankers, or multi-token re-rankers like ColBERT.
- **Generalizability beyond ConTEB**. It was noted that the generalization of results to a wider range of datasets, such as MS-MARCO, would be of interest and more in keeping with existing passage re-rankers.
- **Generalizability beyond Contriever**. It was noted that the generalization of results to a wider range of backbones, beyond Contriever, would be of interest.
- **Impact of hybrid attention**. Multiple reviewers noted that the role of hybrid attention in the performance gains was unclear, motivating some explanation or qualitative analysis.
- **Performance on Football and Insur versus others**. Multiple reviewers noted that the Football and Insur datasets are outliers in terms of performance gains for the method, but the reasons were not clear.

**Reviewer Concerns:**

- **Limitations of document ID embedding**. The authors clarified that the document ID embeddings are relative and local to the retrieved candidates, and thus easily support dynamic corpora. The paper has been updated with this clarification.
  - *Mostly addressed*. This was an important apparent limitation, which the clarification in Section 3.2 seems to satisfactorily.
- **Comparison with document-level and multi-vector re-rankers**. The authors argued that both document-level and multi-vector re-rankers are complementary to the present proposal.
  - *Mostly addressed*. We tend to agree with the authors' view that these ideas are orthogonal to the focus of the submission.
- **Generalizability beyond ConTEB**. The authors noted that the results show good out-of-distribution performance for the proposed method, highlighting there is not overfitting to specific characteristics of ConTEB.
  - *Partially addressed*. The argument about OOD performance on ConTEB is reasonable. However, we understand the concern as being more around the characteristics of the overall benchmark, rather than literally the ConTEB train-test split. Given the ubiquity of MSMARCO for passage re-ranking, a comparison on this dataset seems quite reasonable to ask.
- **Generalizability beyond Contriever**. The authors presented results (Section 4.6) where the E5 backbone is used instead of Contriever, showing good results for the proposed method.
  - *Mostly addressed*. The additional results are a good addition to the paper. It appears that the results are presented somewhat awkwardly inline in Section 4.6, rather than in a table.
- **Impact of hybrid attention**. The authors provided more discussion in Section 4.2 on the hybrid attention module, including an explanation of why it can improve performance. In a nutshell, the argument is that the intra-document attention builds coherent local structures, while the inter-document attention builds global interactions.
  - *Mostly addressed*. The intuition provided is reasonable. In our reading, it seems that the main value of the hybrid attention is the injection of information linking the passages to their underlying documents. Given this, it is unclear whether alternate mechanisms could not capture a similar behavior. For example, what about simply adding a type embedding corresponding to the document? Or, what about appending each passage text with some summary of the document? Such approaches could be simpler to implement than the hybrid attention; if they do not work as well, this is also interesting as it suggests that there is some deeper mechanism at play.
- **Performance on Football and Insur versus others**. The authors explained certain structural properties of these datasets that are amenable to good performance via a cross-passage inference mechanism.
  - *Mostly addressed*.

Further to this, we make a few comments.
- The paper is generally clear, but is verbose in parts. e.g., the long second para on page 2, bullet (iii) on page 3, the last para on page 4.
- Reviewers generally saw the contribution as novel. In our reading, we tend to find the technical contribution somewhat modest -- fundamentally, the proposal involves a particular way of adding document ID information into the passage embedding -- but appreciate the authors have found a successful recipe for an important problem.

**Reviewer Scores:**

- **zHMo**: as the review was generally positive with a score of 8, and the authors clarified the concern around per-document ID embeddings, we think it likely the score would remain the same.
- **dequ**: most of the reviewers' concerns were addressed reasonably well, particularly around the choice of backbone and the apparent need for per-document ID embeddings. We think it plausible that the reviewer would increase their score to 6.
- **VaSb**: the reviewers' concerns around the intuition for the hybrid attention module, and the outlier results on Football/Insur, were addressed reasonably well. The request for runtime against Contriever seems reasonable, but we are unsure if the reviewer had other concerns in mind. We think it possible that the reviewer would increase their score to 6.
- **AjAm**: the authors argued that some of the concerns were best suited for future work. We tend to agree, but are unsure of the extent of the reviewers' concern in them. The authors did provide additional qualitative analysis that could satisfy one of the reviewers' concerns. Overall, we think it plausible that the reviewer would increase their score to 4. We however think it less likely there would be a further increase in score.

---

### Decision · Program_Chairs · 2026-01-26

Accept (Poster)